

# Co-expression clustering across flower development identifies modules for diverse floral forms in *Achimenes* (Gesneriaceae)

Wade R. Roberts[1,2] and Eric H. Roalson[1]

[1] School of Biological Sciences, Washington State University, Pullman, WA, USA
[2] Biological Sciences, University of Arkansas, Fayetteville, AR, USA

## ABSTRACT

**Background:** Genetic pathways involved with flower color and shape are thought to play an important role in the development of flowers associated with different pollination syndromes, such as those associated with bee, butterfly, or hummingbird pollination. Because pollination syndromes are complex traits that are orchestrated by multiple genes and pathways, the gene regulatory networks have not been explored. Gene co-expression networks provide a systems level approach to identify important contributors to floral diversification.

**Methods:** RNA-sequencing was used to assay gene expression across two stages of flower development (an early bud and an intermediate stage) in 10 species of *Achimenes* (Gesneriaceae). Two stage-specific co-expression networks were created from 9,503 orthologs and analyzed to identify module hubs and the network periphery. Module association with bee, butterfly, and hummingbird pollination syndromes was tested using phylogenetic mixed models. The relationship between network connectivity and evolutionary rates ($d_N/d_S$) was tested using linear models.

**Results:** Networks contained 65 and 62 modules that were largely preserved between developmental stages and contained few stage-specific modules. Over a third of the modules in both networks were associated with flower color, shape, and pollination syndrome. Within these modules, several hub nodes were identified that related to the production of anthocyanin and carotenoid pigments and the development of flower shape. Evolutionary rates were decreased in highly connected genes and elevated in peripheral genes.

**Discussion:** This study aids in the understanding of the genetic architecture and network properties underlying the development of floral form and provides valuable candidate modules and genes for future studies.

Corresponding author
Wade R. Roberts, wader@uark.edu

## INTRODUCTION

Flowers display a diverse range of colors, shapes, and sizes, and understanding the ecological and genetic factors contributing to their diversity across angiosperms has long been a major goal in biology. Often this diversity has been attributed to pollinator-mediated selection (*Stebbins, 1970*; *O'Meara et al., 2016*; *Gervasi & Schiestl, 2017*) and the relationship

between a plant and its mode of pollination is considered to be one of the key innovations contributing to angiosperm diversification (*Faegri & Van Der Pijl, 1979*; *Fenster et al., 2004*; *Van Der Niet & Johnson, 2012*; *Barrett, 2013*; *Sauquet & Magallón, 2018*). Most plants evolved to rely on biotic or abiotic means in order to move pollen and ensure reproductive success (*Faegri & Van Der Pijl, 1979*; *Fenster et al., 2004*). These biotic pollinators are often attracted to flowers that contain specific traits, such as red flowers that provide high nectar rewards for bird visitors (*Cronk & Ojeda, 2008*) or the ultraviolet markings on some flowers that attract bee visitors (*Papiorek et al., 2016*). A wide range of floral traits are thought to contribute to successful pollination, such as color (*Sletvold et al., 2016*), odor (*Piechowski, Dötterl & Gottsberger, 2010*), nectar composition (*Amorim, Galetto & Sazima, 2013*), and time of flowering (*Cortés-Flores et al., 2017*). The genetic basis for these and other traits and their role in floral and pollination syndrome divergence has been examined extensively in model systems, particularly in *Mimulus* (*Bradshaw et al., 1998*; *Yuan et al., 2016*) and *Petunia* (*Hoballah et al., 2007*; *Hermann et al., 2015*), and more recently in many non-model systems (*Wessinger, Hileman & Rausher, 2014*; *Alexandre et al., 2015*).

Among the most widely used large-scale experimental approaches to investigate genome function are transcriptome analyses (*Pickrell et al., 2010*; *Raherison et al., 2015*), particularly in non-model organisms where no reference genome or functional genomics data exists. Changes in expression often result from the combinatorial action of genetic regulatory pathways orchestrating development and responses to environmental stimuli. Therefore, the transcriptome can be viewed as a link between the genotype and phenotype and may be acted upon through selection (*Romero, Ruvinsky & Gilad, 2012*; *Prasad et al., 2013*). As changes in gene expression may underlie many of the phenotypic changes between species (*Brawand et al., 2011*; *Romero, Ruvinsky & Gilad, 2012*; *Uebbing et al., 2016*), studying the transcriptome may shed light on important pathways and targets of selection. Phenotypic changes can frequently arise through functional changes in conserved developmental pathways among closely related species. For example, changes in enzyme function and transcriptional regulation of the anthocyanin pathway has been implicated frequently across angiosperms in the evolution of red flowers (*Des Marais & Rausher, 2010*; *Smith & Rausher, 2011*). It is now recognized that most genes act as members of biological pathways or of co-regulated modules (*Hollender et al., 2014*; *Ma et al., 2018*).

Here, we undertake a comparative study of gene expression during flower development in the genus *Achimenes*. This group is a member of the diverse African violet family (Gesneriaceae) and distributed throughout Mexico and Central America. *Achimenes* is a young lineage (*c.* 7–12 Mya; *Roalson & Roberts, 2016*) known for its floral diversity (Fig. 1), a feature thought to be associated with speciation (*Ramírez Roa, 1987*; *Roalson, Skog & Zimmer, 2003*). Four pollination syndromes are found in *Achimenes*, including melittophily (bees), psychophily (butterflies), euglossophily (female euglossine bees) and ornithophily (hummingbirds) (Fig. 2). These syndromes have traditionally been defined on the basis of flower color and flower shape (Fig. 2; *Ramírez Roa, 1987*) and recently through pollinator observations (*Martén-Rodríguez et al., 2015*; *Ramírez-Aguirre et al., 2019*).

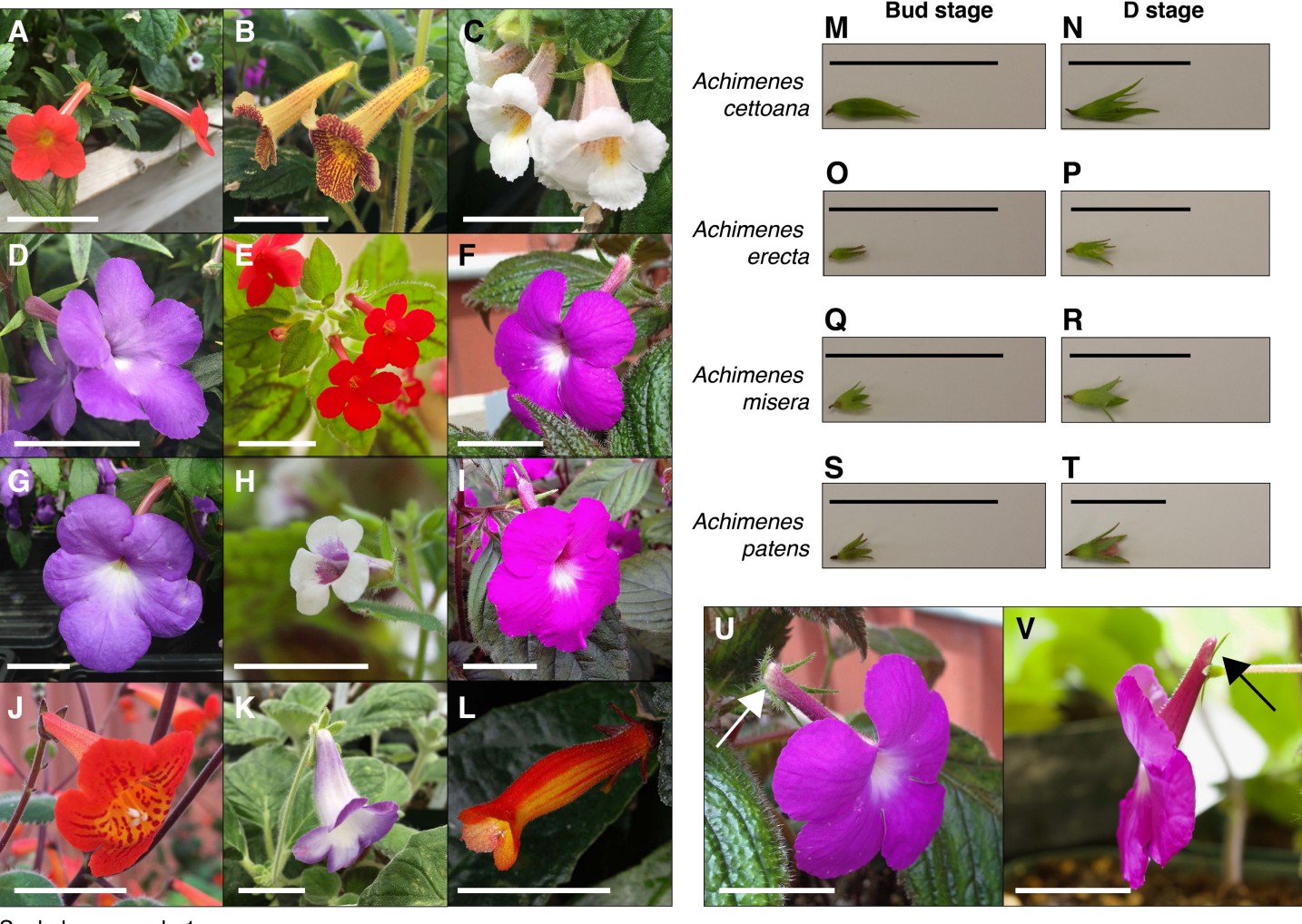

Scale bars equals 1 cm.

**Figure 1** ***Achimenes* flowers and the sampled developmental stages.** Flowers of the twelve sampled species: (A) *A. admirabilis*, (B) *A. antirrhina*, (C) *A. candida*, (D) *A. cettoana*, (E) *A. erecta*, (F) *A. grandiflora*, (G) *A. longiflora*, (H) *A. misera*, (I) *A. patens*, (J) *A. pedunculata*, (K) *E. verticillata* and (L) *G. cuneifolia*. Sampled Bud and D stage flowers for (M and N) *A. cettoana*, (O and P) *A. erecta*, (Q and R) *A. misera* and (S and T) *A. patens*. Corolla spurs found in (U) *A. grandiflora* and (V) *A. patens* are indicated with arrows. Scale bars equal 1 cm. Abbreviations: *A, Achimenes*; *E, Eucodonia*; *G, Gesneria*.

Currently there have been pollinator observations made for four *Achimenes* species: *A. antirrhina* (hummingbirds, *Amazilia beryllina*), *A. flava* (bees, Anthophoridae), *A. obscura* (bees, *Trigona fulviventris*) and *A. patens* (unidentified butterflies) (*Martén-Rodríguez et al., 2015*; *Ramírez-Aguirre et al., 2019*). Based on these observations and the floral traits of these species (Fig. 2), we can hypothesize the likely pollinators for the other *Achimenes* species (Fig. 2; Table 1). Repeated origins and transitions between these pollination syndromes have been hypothesized in *Achimenes* (*Roalson, Skog & Zimmer, 2003*), making this small lineage an attractive system to understand the genetic and ecological factors underlying floral diversification.

In this study, we aimed to characterize and compare patterns of gene co-expression during flower development across 10 *Achimenes* species. So far, few studies have

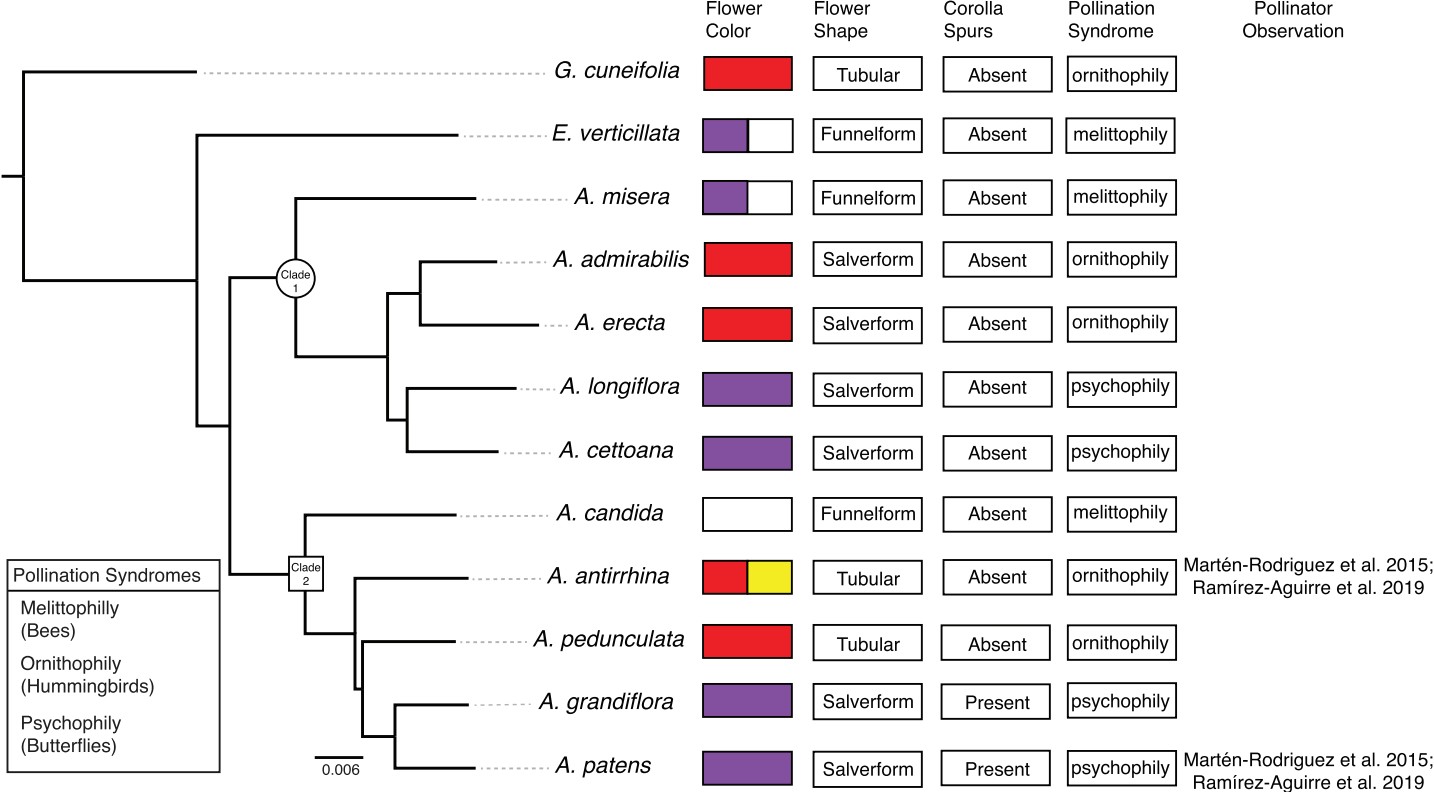

**Figure 2 Phylogenetic relationships and floral traits in *Achimenes*.** Phylogeny of *Achimenes* adapted from *Roberts & Roalson (2018)*. Clade 1 and Clade 2 (sensu *Roalson, Skog & Zimmer (2003)*) are indicated on the phylogeny with a circle and box, respectively. Flower traits for each species are shown at the tips of the phylogeny, including flower color, flower shape, presence of corolla spurs, pollination syndrome and pollinator observations.

**Table 1 Definition of pollination syndromes in *Achimenes*, *Eucodonia*, and *Gesneria*.**

| Pollination syndrome | Primary flower color | Flower shape | Presence of corolla spurs | Species |
|---|---|---|---|---|
| Melittophily (bees) | White or Purple with yellow or purple spotted throat | Funnelform | No | *Achimenes candida* |
| | | | | *Achimenes misera* |
| | | | | *Eucodonia verticillata* |
| Ornithophily (hummingbirds) | Red and yellow | Tubular, Salverform | No | *Achimenes admirabilis* |
| | | | | *Achimenes antirrhina* |
| | | | | *Achimenes erecta* |
| | | | | *Achimenes pedunculata* |
| | | | | *Gesneria cuneifolia* |
| Psychophily (butterflies) | Purple | Salverform | Yes | *Achimenes cettoana* |
| | | | | *Achimenes grandiflora* |
| | | | | *Achimenes longiflora* |
| | | | | *Achimenes patens* |

investigated the genetic basis of floral development in the Gesneriaceae family (*Alexandre et al., 2015*; *Roberts & Roalson, 2017*). Given that bee, butterfly, and hummingbird pollination syndromes evolved in parallel across *Achimenes*, we hypothesized that: (1) distinct sets of co-expressed genes would correlate to each syndrome during flower development, and (2) genes in pathways involved with flower color and shape may be central in each network during flower development. Using a comparative transcriptome approach, we clustered a shared set of 9,503 orthologs into two different co-expression networks, each corresponding to two stages of flower development: an early bud stage and an intermediate stage prior to anthesis. This strategy allowed us to compare gene co-expression patterns temporally. We compared these networks and tested whether gene co-expression clusters correlated with different floral traits (pollination syndrome, flower color, flower shape and corolla spurs). We also tested whether network properties, such as connectivity and expression, influence evolutionary constraints. Our results allowed us to quantify the extent of shared gene co-expression among closely related species and to identify pathways and genes that may be associated with the repeated evolution of flower types underlying pollinator specificity.

## MATERIALS AND METHODS

### Plant materials

Ten of the 26 currently recognized species in *Achimenes* were sampled: five from Clade 1 and five from Clade 2 (Fig. 2). These species were chosen based on their hypothesized close relationships and their diversity in floral form (Fig. 1; *Roalson, Skog & Zimmer, 2003*). On the basis of previous molecular studies (*Roalson & Roberts, 2016*), two species were chosen as outgroups to represent related lineages: *Eucodonia verticillata* and *Gesneria cuneifolia* (Figs. 1 and 2). All selected species were grown in standard greenhouse conditions at Washington State University, under 16-h days, 24–27 °C and 80–85% humidity.

Whole flower buds from two stages of flower development were sampled for each species: an immature bud stage (Bud stage; Fig. 1) and an intermediate stage before anthesis (D stage; Fig. 1). Our definitions of the floral developmental stages in *Achimenes* were adapted from the stages determined for *Antirrhinum majus*, which uses a morphological and temporal framework (*Vincent & Coen, 2004*). The Bud stage represents the smallest reproductive bud distinguishable from vegetative buds, having lengths between 1 and 2 mm (Fig. 1). The appearance is marked by the emergence of the floral primordia, no accumulation of anthocyanin pigments and no elongation or growth of corolla tissue cells. The D stage represents a halfway point between the Bud and a pre-anthesis flower, where the length is about half the length of a fully developed flower (Fig. 1). Its appearance is marked by the accumulation of anthocyanins in the corolla tissue, the cells have elongated in the corolla tube and trichomes have emerged. Corolla spurs are found only in *A. grandiflora* and *A. patens*, and the spur begins development during the D stage as an outward growth of the corolla in the opposite direction from the tube opening (Fig. 1).

Each stage was sampled with two or three biological replicates, contributing four or six samples for each species. The bud stage was chosen to represent a baseline level of gene expression when floral phenotypes are largely similar between species that could then be compared to the intermediate stage where floral phenotypes diverge and many important pathways involved in flower development and pigmentation show increased expression levels (*Roberts & Roalson, 2017*). All tissues were immediately frozen into liquid nitrogen and stored at −80 °C.

## RNA extraction and RNA-seq library construction

Total RNA was extracted from frozen tissue using the RNEasy Plant Kit (Qiagen, Hilden, Germany) and ribosomal depleted RNA samples were prepared using the RiboMinus Plant Kit (Thermo Fisher Scientific, Waltham, MA, USA). Stranded libraries were prepared using the NEBNext Ultra Directional RNA Library Kit (New England Biolabs, Ipswich, MA, USA), barcoded, and pooled based on nanomolar concentrations. Library quality and quantity were assessed using the Qubit (dsDNA HS assay; Thermo Fisher Scientific, Waltham, MA, USA) and BioAnalyzer 2100 (Agilent, Santa Clara, CA, USA). The library pool was sequenced across four lanes of an Illumina HiSeq2500 for paired-end 101 bp reads at the Genomics Core Lab at Washington State University, Spokane. All new sequence data were deposited in the NCBI Sequence Read Archive (BioProject: PRJNA435759).

Single replicate libraries for four of the species in the current study (*A. cettoana*, *A. erecta*, *A. misera* and *A. patens*) were previously sequenced for the same two timepoints (Bud and D stages) (*Roberts & Roalson, 2017*). These samples were combined with the newly generated data to assembly de novo transcriptomes and perform co-expression clustering. This data was deposited in the NCBI Sequence Read Archive (BioProject: PRJNA340450).

## Transcriptome assembly

Per base quality, adapter contamination, and per base sequence content of raw reads was assessed with FastQC tools (http://bioinformatics.babraham.ac.uk/projects/fastqc). One library (AT6) returned poor yields and low-quality reads and was excluded from further analyses. Trimmomatic v.0.36 (*Bolger, Lohse & Usadel, 2014*) was then used to remove contaminating adapter sequences and dynamically trim low-quality bases (ILLUMINACLIP:TruSeq3-PE-2.fa:2:30:10 HEADCROP:13 LEADING:3 TRAILING:3 SLIDINGWINDOW:4:15 MINLEN:50). Trimmed reads derived from an individual species were combined and de novo assembled into contigs using Trinity (–SS_lib_type RF; *Grabherr et al., 2011*; *Haas et al., 2013*), generating a reference transcriptome for each species. Sequence redundancy was reduced with CDHIT-EST (-c 0.99 -n 5; *Li & Godzik, 2006*). Open reading frames were then predicted for each transcriptome assembly with TransDecoder (*Haas et al., 2013*) using a BLASTp search against the SwissProt database (www.uniprot.org) to maximize sensitivity. Finally, transcriptome assembly completeness was assessed using BUSCO (*Simão et al., 2015*) against a set of single-copy plant orthologs (embryophyte_odb10 dataset).

## Orthogroup inference

To help facilitate orthogroup inference, transcriptome assemblies from flowers in *Sinningia eumorpha* and *S. magnifica* were downloaded and combined with our assemblies (https://doi.org/10.5061/dryad.4r5p1; *Serrano-Serrano et al., 2017a*). *Sinningia* is closely related to our focal lineage in the Neotropical Gesneriaceae (*Roalson & Roberts, 2016*). To identify orthogroup sequences across all 14 species, we used a modified version of the approach described in *Yang & Smith (2014)*. All peptide sequences were clustered using an all-by-all BLASTp search performed using DIAMOND (–evalue 1e−6; *Buchfink, Xie & Huson, 2015*) and results were clustered using MCL v14-137 (-I 1.4; *Enright, Van Dongen & Ouzounis, 2002*). Homolog clusters were aligned using MAFFT v7.271 (–genafpair –maxiterate 1000; *Katoh & Standley, 2013*) and alignments were trimmed using Phyutility v2.7.1 (-clean 0.1; *Smith & Dunn, 2008*). Codon alignments were then produced for each cluster using PAL2NAL v14 (*Suyama, Torrents & Bork, 2006*) and maximum-likelihood trees were constructed using FastTree v2.1.8 (*Price, Dehal & Arkin, 2010*) using the GTR model. Trees were then trimmed to exclude branches >10 times longer than its sister or longer than 0.2 substitutions per site. Final homolog group alignments were created from the trimmed trees using MAFFT and used for final homolog tree inference using RAxML v8.2.9 (*Stamatakis, 2014*) and the GTRGAMMA model. Orthogroups were inferred from the final homolog trees using the rooted ingroup (RT) method of *Yang & Smith (2014)*, using *S. eumorpha* and *S. magnifica* as outgroups. Homolog trees were trimmed to include at least 2 ingroup species, resulting in 83,595 orthogroups.

## Quantifying gene expression

Trimmed reads for each sample were aligned to the corresponding transcriptome assembly to quantify expression using default parameters in Kallisto v0.43.0 (*Bray et al., 2016*). The resulting counts for each contig were matched with the inferred orthogroups derived above, creating two separate matrices for the Bud and D stage samples. In the resulting gene expression matrices, rows corresponded to orthogroups and columns corresponded to individual samples. Counts for the Bud and D stage samples were separately transformed using variance-stabilizing transformation implemented in the Bioconductor package DESeq2 (*Love, Huber & Anders, 2014*) and quantile normalized using the Bioconductor package preprocessCore (*Bolstad, 2018*). Lastly, counts for both stages were separately corrected for confounding effects, such as batch or species effects, using principal component regression with the "sva_network" function in the Bioconductor package sva (*Parsana et al., 2019*).

## Co-expression networks and identification of modules

The Bud and D stage RNAseq data was clustered into gene co-expression networks using the R package WGCNA (*Langfelder & Horvath, 2008*). Prior to network construction, orthogroups were filtered to remove any with >50% missing values or <0.3 expression variance. In total, 9,503 orthogroups common to both Bud and D stage data were used to construct separate networks for each stage. Parameters for the Bud stage network were as follows: power = 18, networkType = "signed", corType = "bicor", maxPOutliers = 0.05,

TOMType = "signed", deepSplit = 2, mergeCutHeight = 0.25. For the D stage network, the following parameters were used: power = 16, networkType = "signed", corType = "bicor", maxPOutliers = 0.05, TOMType = "signed", deepSplit = 2, mergeCutHeight = 0.25. Soft-thresholding powers for each network ("power" parameter) were chosen as the lowest value such that the scale free topology model fit ($R^2$) was ≥0.9. Signed networks consider positively correlated nodes connected, while unsigned networks consider both positively and negatively correlated nodes connected (*Van Dam et al., 2018*). We used signed networks here ("networkType" and "TOMType" parameters) because this type is considered more robust to biological functions with more specific expression patterns (*Mason et al., 2009*; *Song, Langfelder & Horvath, 2012*). Unsigned networks can capture only strong correlations and many negative regulatory relationships in biological systems are weak or moderate (*Ritchie et al., 2009*). Biweight midcorrelation ("corType" parameter) was used as this method is often more robust than Pearson correlation and more powerful than Spearman correlation (*Song, Langfelder & Horvath, 2012*). All other WGCNA parameters were kept at their default values.

Module eigengenes and orthogroup connectivity were calculated in each network separately using the "moduleEigengenes" and "intramodularConnectivity" functions in WGCNA, respectively. The module eigengene is defined as the first principal component of a module and represents the gene expression profile of a sample within a module (*Langfelder & Horvath, 2008*). Connectivity refers to the sum of connection strengths a node has to other network nodes.

Module hub genes (defined as the most highly connected nodes within a module) were identified in both networks based on module membership (kME) scores >0.9, calculated using the "signedKME" function of WGCNA. kME is defined as the correlation between the expression levels of a gene and the module eigengene (*Horvath & Dong, 2008*). Additionally, using connectivity measures, we defined peripheral genes in each network as the lowest 10% of connected nodes.

We tested whether Bud stage modules were preserved in the D stage and vice versa using module preservation statistics in WGCNA (*Langfelder et al., 2011*). Median Rank and Zsummary statistics were computed using the "modulePreservation" function of WGCNA, using 200 permutations (*Langfelder et al., 2011*). Median rank statistics were compared to the "gold" module, which consists of 1,000 randomly selected orthogroups. Modules with a lower median rank exhibit stronger preservation than modules with a higher median rank. Zsummary scores >10 indicate strong evidence of preservation and scores <2 indicate no evidence of preservation.

## Phylogeny

We previously inferred a phylogeny for the 12 sampled species using 1,306 single-copy orthologs identified from the same transcriptome dataset used here (*Roberts & Roalson, 2018*). For comparative analyses of module-trait relationships, we randomly sampled 50 single-copy ortholog gene trees and rescaled branch lengths to be proportional to time (ultrametric) using the "chronos" function in the R package ape (*Paradis, Claude & Strimmer, 2004*). Bootstrap support was 100 for nearly every branch in the

*Roberts & Roalson (2018)* phylogeny, therefore we chose to use randomly sampled ortholog trees to account for phylogenetic uncertainty.

## Module and trait relationship

We correlated external traits (e.g., red flower color) with the module eigengenes using a phylogenetic Markov Chain Monte Carlo method, implemented in the R packages MCMCglmm (*Hadfield, 2010*) and mulTree (*Guillerme & Healy, 2014*). These analyses were performed individually for each module in both the Bud and D stage networks. Three floral traits and pollination syndrome were coded: primary flower color (purple, red, white and yellow), flower shape (funnelform, salverform and tubular), corolla spur (absent and present), and syndrome (bee, butterfly and hummingbird). We fit a multivariate mixed model with a non-informative prior, where the shape and scale parameters equal 0.001, and residual variance was fixed. Analyses were performed over the 50 randomly selected ortholog trees to account for phylogenetic uncertainty. A phylogenetic model was used to account for any interspecific non-dependency in the dataset by using a random effects structure that incorporated a phylogenetic tree model. Floral traits were set as categorical response variables and the module eigengenes were set as the predictor variable. For each tree, two MCMC chains were run for 250 k generations and discarding the first 50 k as burn-in. This process was run individually over each module in both the Bud and D stages. We checked for convergence between model chains using the Gelman–Rubin statistic (*Gelman & Rubin, 1992*). Potential scale reduction values were all less than 1.1 and effective sample sizes for all fixed effects were greater than 400. We considered fixed effects to be statistically significant when the probabilities in the 95% credible region did not include zero (Figs. S3–S6).

## Evolutionary rate analysis

Codon alignments of the filtered orthogroups ($n = 9,503$) were produced using PAL2NAL v14 (*Suyama, Torrents & Bork, 2006*). Subsequent alignments and corresponding gene trees were used as input to the codeml package in PAML v4.9 (*Yang, 2007*) to estimate $d_N/d_S$ (omega, ω). $d_N/d_S$ for each orthogroup was estimated using a one-ratio model (model = 0, NSsites = 0), providing a single estimate for each orthogroup to match other single orthogroup metrics, such as connectivity. If $d_N/d_S$ values were exceptionally large (=999) because of zero synonymous differences, those values were removed from the dataset.

First, we performed linear regression (LM) to test the effect of orthogroup connectivity, average expression levels, and their interactions (explanatory variables) on the estimated $d_N/d_S$ values (main effect), LM = $d_N/d_S$ ~ connectivity * expression. This was performed using the lm function in R with 10,000 bootstrap pseudoreplicates. Second, we ran two-sample *t*-tests with 10,000 permutations to test whether hub nodes and periphery nodes in each network had lower (for hubs) or higher (for periphery) $d_N/d_S$ values than all background nodes. Third, we fit a LM with 10,000 bootstrap pseudoreplicates to test whether modules associated with a pollination syndrome (bee, butterfly, or hummingbird; see "Results" below) had increased $d_N/d_S$ values, LM = $d_N/d_S$ ~ syndrome.
For all statistical analyses, $d_N/d_S$, connectivity, and expression values were log transformed to reach a normal distribution before being processed. Analyses were performed separately for each network using the separately estimated connectivity and expression levels.

## Functional annotation and gene ontology enrichment analysis

Filtered orthogroups ($n$ = 9,503) were annotated based on a search against all green plant proteins in the SwissProt database (release 2018_07; *The UniProt Consortium, 2017*) using Diamond BLASTp (—evalue 1e−6). Results were used to assign annotations and gene ontology (GO) terms. GO enrichment analyses were performed using the R package TopGO and the default "weight01" algorithm with the recommended cutoff of $p < 0.05$ (*Alexa, Rahnenführer & Lengauer, 2006*). Enriched GO term redundancy was removed and summarized using the semantic similarity measures implemented in the REVIGO web server (*Supek et al., 2011*).

## Data and code availability

Raw sequencing reads are available from the NCBI Sequence Read Archive under BioProjects: PRJNA435759 and PRJNA340450. Data and code for all analyses can found at Zenodo: DOI 10.5281/zenodo.3517231.

# RESULTS

## Transcriptome assembly

Twelve transcriptomes were sequenced (72 libraries) and assembled in 12 species after sampling across two floral developmental stages (Table S1). The transcriptomes had an average assembly length of 244.9 Mb (±33.9 Mb), an average of 217,510 (±47 k) transcripts, and an average N50 length of 1,988 bp (±306 bp). Between 22,942 and 33,501 putative genes were detected with 2.34 (±0.35) putative isoforms. BUSCO identified an average of 87% of the complete, conserved single-copy plant orthologs (Table S1). Our focus in our comparative study was on the presence of conserved orthologs, and not on the presence of taxonomically restricted genes, which are not likely preserved across species.

## Constructing the co-expression network for 12 species

We identified 83,595 orthologous clusters (orthogroups), each containing at least two species. After removing 74,092 orthogroups with too many missing values (>50% missing) or low expression variance (<0.3 variance) and keeping common orthogroups shared between the Bud and D stages, 9,503 orthogroups were used to construct two co-expression networks using the weighted gene correlation network analysis (WGCNA) approach. Modules of co-expressed orthogroups are inferred using the expression profiles of each sample regardless of species. Clustering identified 65 and 62 co-expression modules in the Bud and D stage networks, respectively (Fig. 3; Table S2). Module size in the Bud stage ranged from 22 (module ME63) to 592 (ME1) orthogroups (mean = 146) and ranged from 21 (ME61) to 704 (ME1) orthogroups (mean = 153) in the D stage. Out of
A

**Cluster Dendrogram**

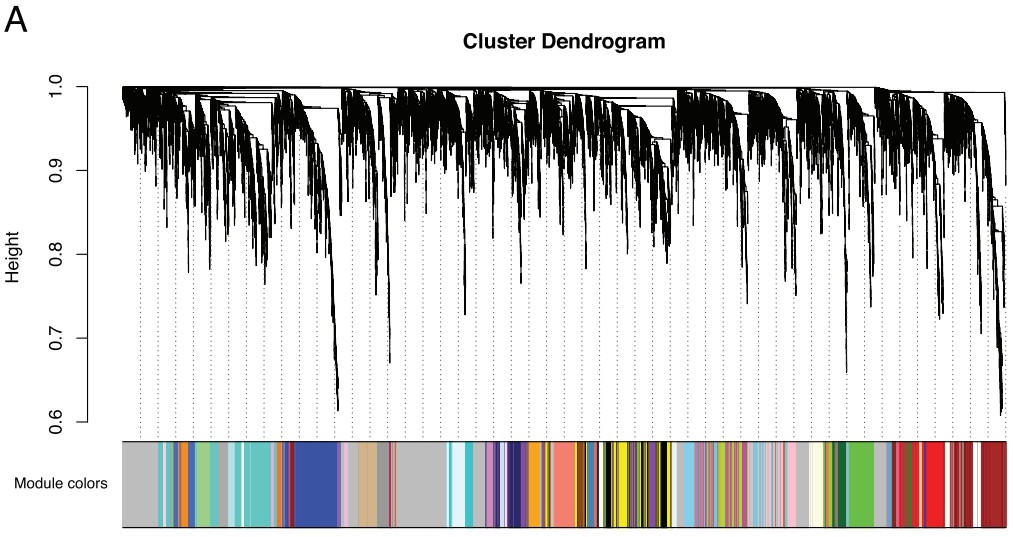

B

**Cluster Dendrogram**

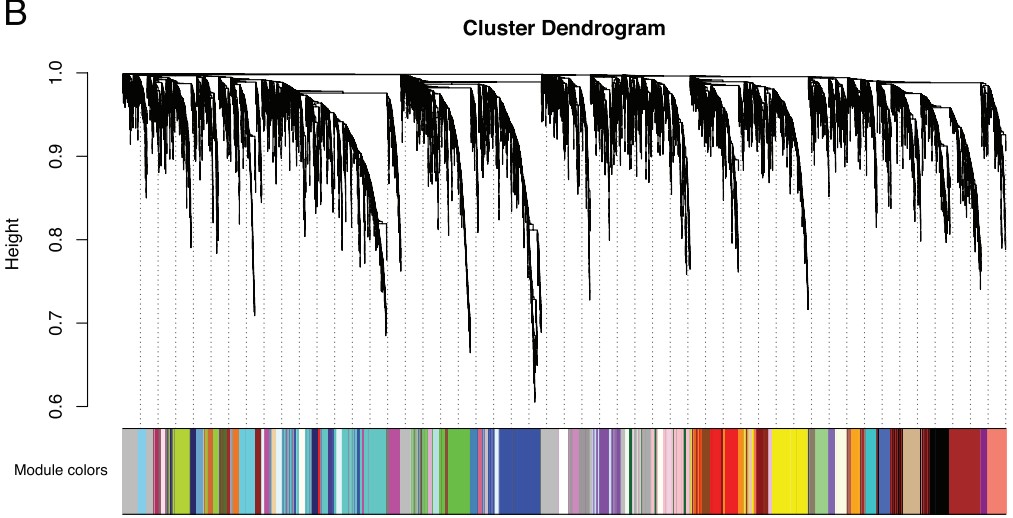

**Figure 3 Co-expression clustering of the Bud and D stages.** Hierarchical clustering by WGCNA showing co-expressed modules for (A) the Bud stage and (B) the D stage. Each of the 9,503 orthogroups were assigned to 65 and 62 modules, respectively, and the color rows underneath show the module assignment.

the 9,503 orthogroups, 7,845 (83%) and 8,735 (92%) were assigned to modules in each network.

The remaining unassigned genes from each network that were not placed into a well-defined module were assigned to the ME0 module by WGCNA. Despite being unplaced, these orthogroups may still be functionally relevant within each network. Unassigned genes in the ME0 module were enriched for many processes, including transcription (Bud, $n = 173$ and D, $n = 84$) and organ development (D, $n = 55$), among other functions (Tables S3 and S4).

We tested whether modules in the Bud stage were preserved in the D stage, and vice versa. Overall, preservation of modules between the Bud and D stage networks was high.
# Correspondence of Bud and D stage modules

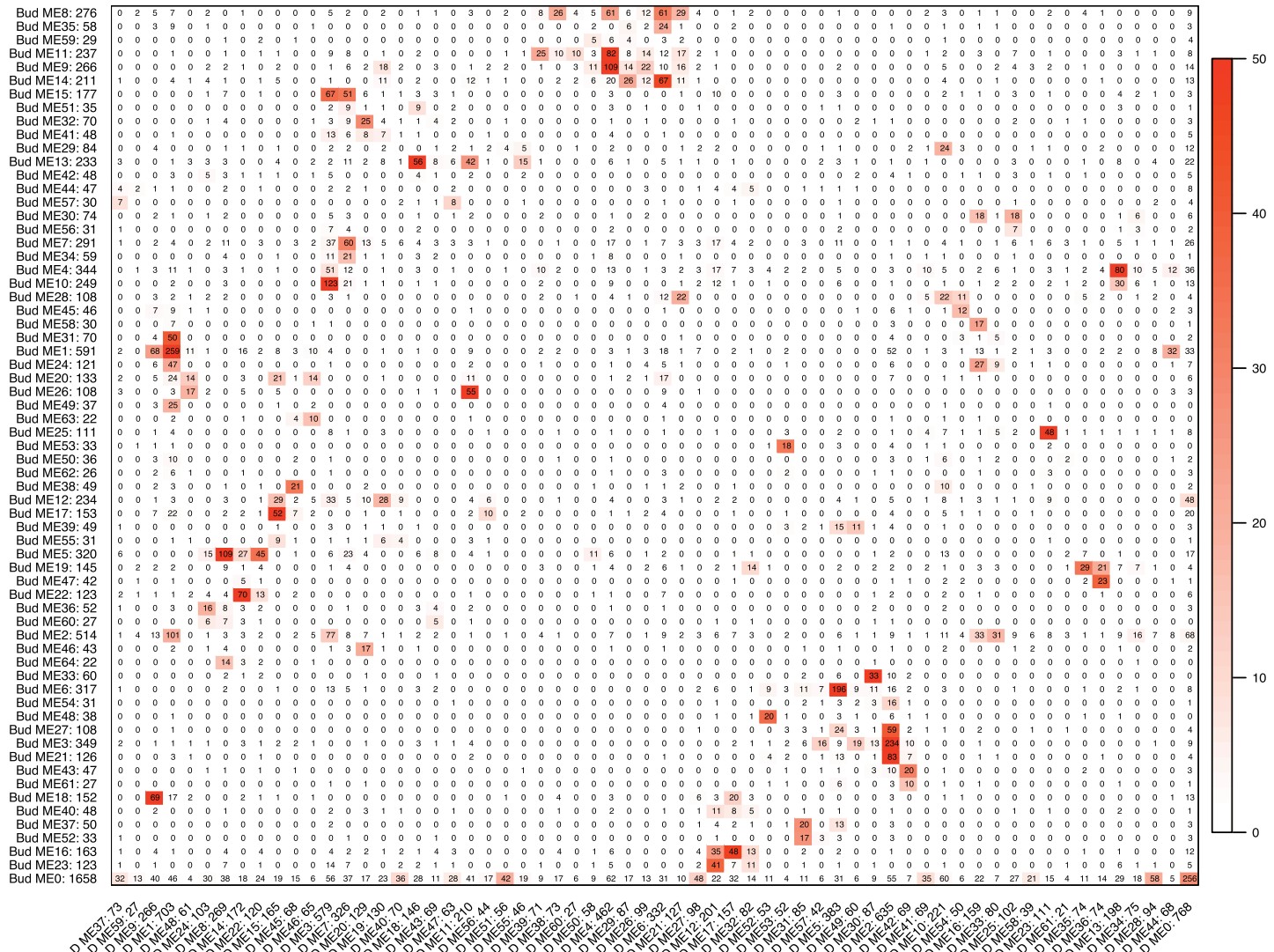

**Figure 4 Correspondence of the Bud and D stage networks.** Each row corresponds to one Bud stage module (labeled by module name and module size), and each column corresponds to one D stage module (labeled by module name and module size). Numbers in the table indicate gene counts in the intersection of the corresponding modules. Coloring of the table encodes –log(p), with *p* being the Fisher's exact test *p*-value for the overlap of the two modules. The stronger the red color, the more significant the overlap.

Eight of the 65 Bud stage modules (ME46, ME62, ME44, ME57, ME42, ME41, ME55 and ME40) were not strongly preserved in the D stage, while only 1 of the 62 D stage modules (ME59) was not strongly preserved in the Bud stage (Table S5). These patterns suggest that these non-preserved modules may play unique roles within their network. Among the eight Bud stage-specific modules were many orthogroups related to primary cell growth, meristem identity and differentiation, stamen development and flowering (Table S3). In the single D stage-specific module were many orthogroups related to methylation, flowering, and carotenoid biosynthesis (Table S4). Despite the relatively high preservation,

correspondence between module membership in the Bud and D stages was moderate (Fig. 4). Orthogroup module membership tended to shift between each network. For example, most members of module ME15 in the Bud stage are transferred to modules ME3 and ME7 in the D stage (Fig. 4).

We defined network hubs (the most highly connected nodes within a module) with module membership (kME) scores >0.9. Using this definition, all modules in each network contained at least one hub, with the percentage of hubs ranging from 1% to 24% of orthogroups in a module (Figs. 5 and 6; Table S6). Of the 632 and 691 hubs identified in the Bud and D stages, a significant portion (110) were shared between networks (Fisher's Exact Test, odds ratio = 3.02, $P < 0.001$). Functional enrichment of hubs in the Bud stage revealed that many were involved in the regulation of development and metabolic processes ($n = 40$), transport and protein localization ($n = 39$), biosynthetic processes ($n = 19$), and DNA methylation ($n = 7$), consistent with the notion that hubs usually play important roles in integrating other genes during network evolution (*Ravasz et al., 2002*) (Table 2). In the D stage, hubs were enriched for functions related to translation ($n = 33$), transport ($n = 9$), ethylene and abscisic acid signaling pathways ($n = 17$), carotenoid and anthocyanin biosynthetic processes ($n = 11$), and the regulation of gene expression ($n = 20$), among others (Table 2).

Nodes in the periphery of each network were loosely defined by considering them to be the lowest 10% connected (connectivity$_{Bud}$ < 2.824, connectivity$_D$ < 4.651). These definitions identified 951 periphery nodes (Table S7), of which 301 were peripheral in both networks (Fisher's Exact Test, odds ratio = 5.67, $P < 0.001$). These periphery nodes in the Bud stage were enriched for functions related to reproductive structure development ($n = 75$), transport and localization ($n = 107$), and signaling ($n = 389$), among others (Table 3). In the D stage, the periphery was enriched for functions related to the regulation of flower development ($n = 64$), the regulation of flowering time ($n = 18$), the regulation of flavonoid biosynthesis ($n = 7$), and cell fate specification ($n = 12$), among others (Table 3). Greater than expected numbers of peripheral nodes in the D stage were related to regulation (Fisher's exact test, odds ratio = 1.191, $p = 0.02752$) and transcription (Fisher's exact test, odds ratio = 1.370, $p = 0.001549$). In contrast, there were no differences in the expected number of peripheral nodes related to regulation (Fisher's exact test, odds ratio = 0.955, $p = 0.7044$) or transcription (Fisher's exact test, odds ratio = 1.047, $p = 0.3549$) in the Bud stage.

## Modules associated with floral traits

We calculated module eigengenes, single values which represent the gene expression profiles of a sample within a module (Figs. S1 and S2). The extent of module involvement in various biological processes can be tested by correlating eigengenes with external traits, such as flower color. We tested the relationship between the module eigengenes and the four floral traits (pollination syndrome (bee, butterfly, hummingbird), flower color (purple, red, yellow and white), flower shape (funnelform, salverform and tubular), and corolla spurs (absence and presence)), while controlling for any phylogenetic bias in the dataset using a phylogenetic mixed model (Figs. 5 and 6; Figs. S3–S6). We also examined

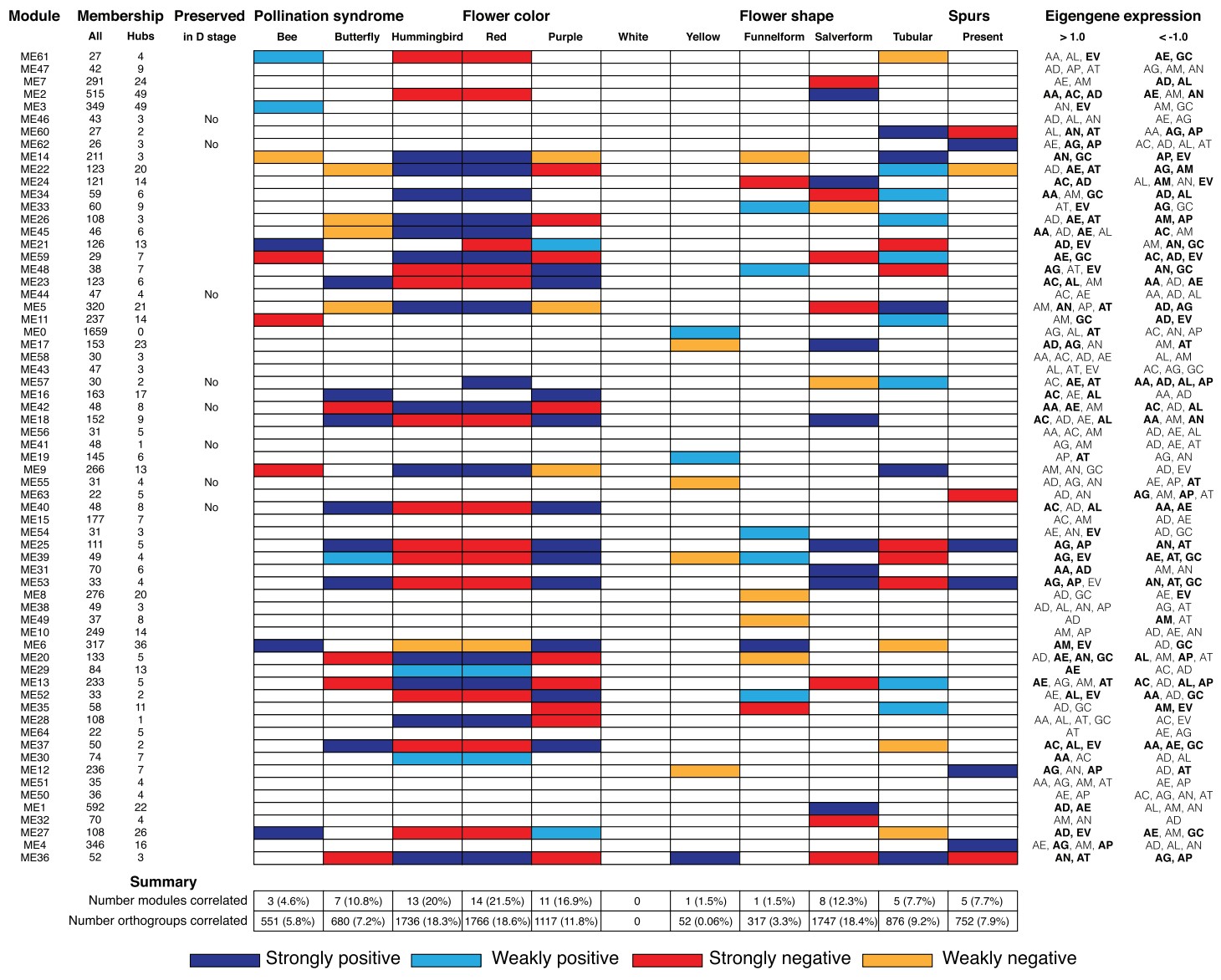

**Figure 5  Summary of Bud stage network module association to floral traits and pollination syndrome.** Each row represents an individual module in the Bud stage network and shows the total size (column All), the number of hubs (column Hubs), and whether the module is preserved in the D stage. Each column in the central table represents the floral traits (Pollination Syndrome, Flower color, Flower shape and Spurs) that were tested for an association to module eigengenes using MCMCglmm. The eigengene expression is indicated for each module in which all three replicates from an individual species had >1.0 or <−1.0 expression. An association was considered strong when both MCMCglmm results were significant and the eigengene expression was >1.0 or <−1.0 in multiple species that shared the significant trait. An association was considered weak when MCMCglmm results were significant but the eigengene expression was >1.0 or <−1.0 in a single species or no species that shared the significant trait. Eigengene expression for species that share traits to those found significant by MCMCglmm are in bold.

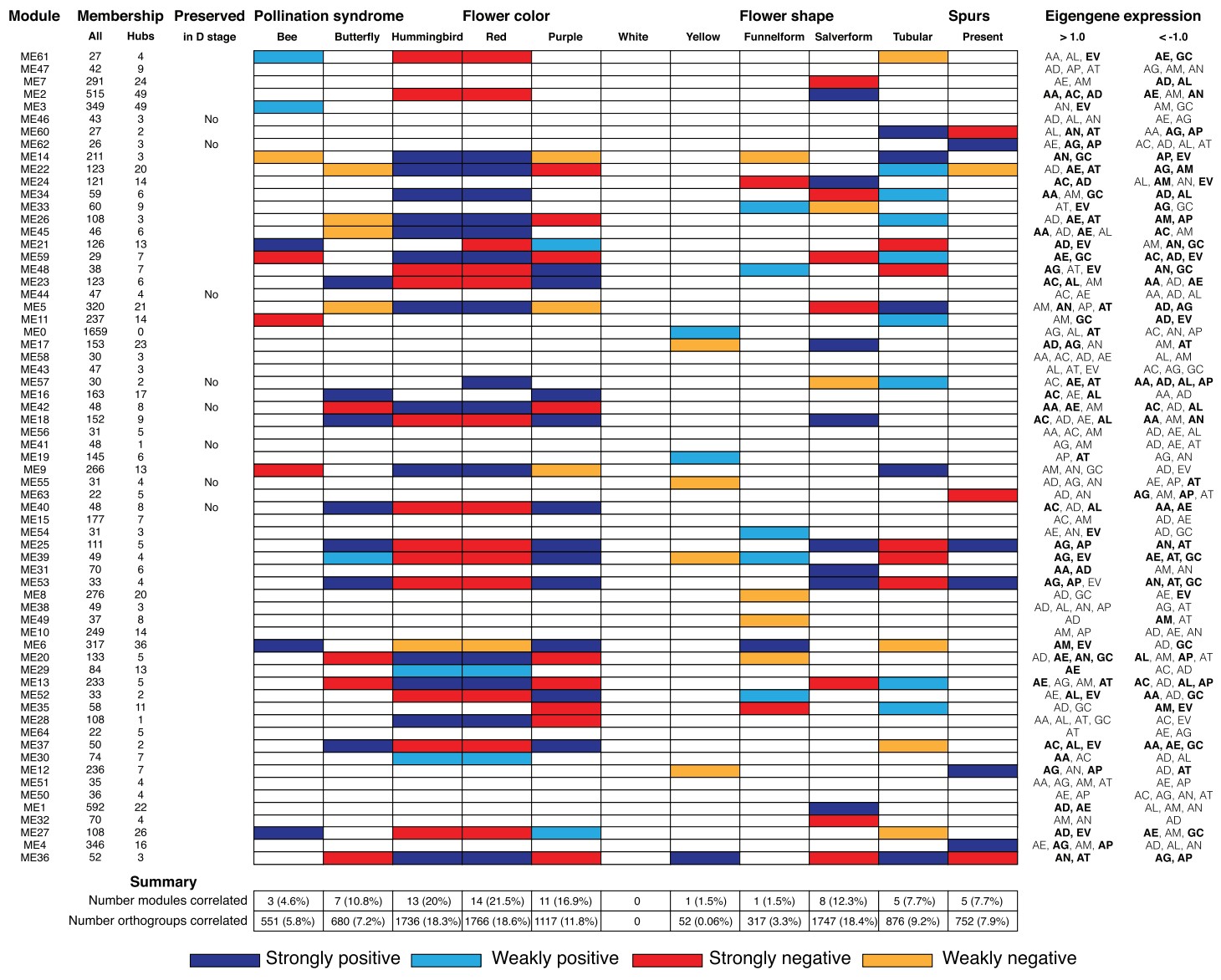

whether eigengene values (>0.1 or <−0.1) in single species contributed to a strong correlation or whether eigengene values (>0.1 or <−0.1) in multiple species contributed to a strong correlation (Figs. 5 and 6; Figs. S3–S6). We considered there to be strong evidence for trait association when a significant correlation from MCMCglmm was coupled with

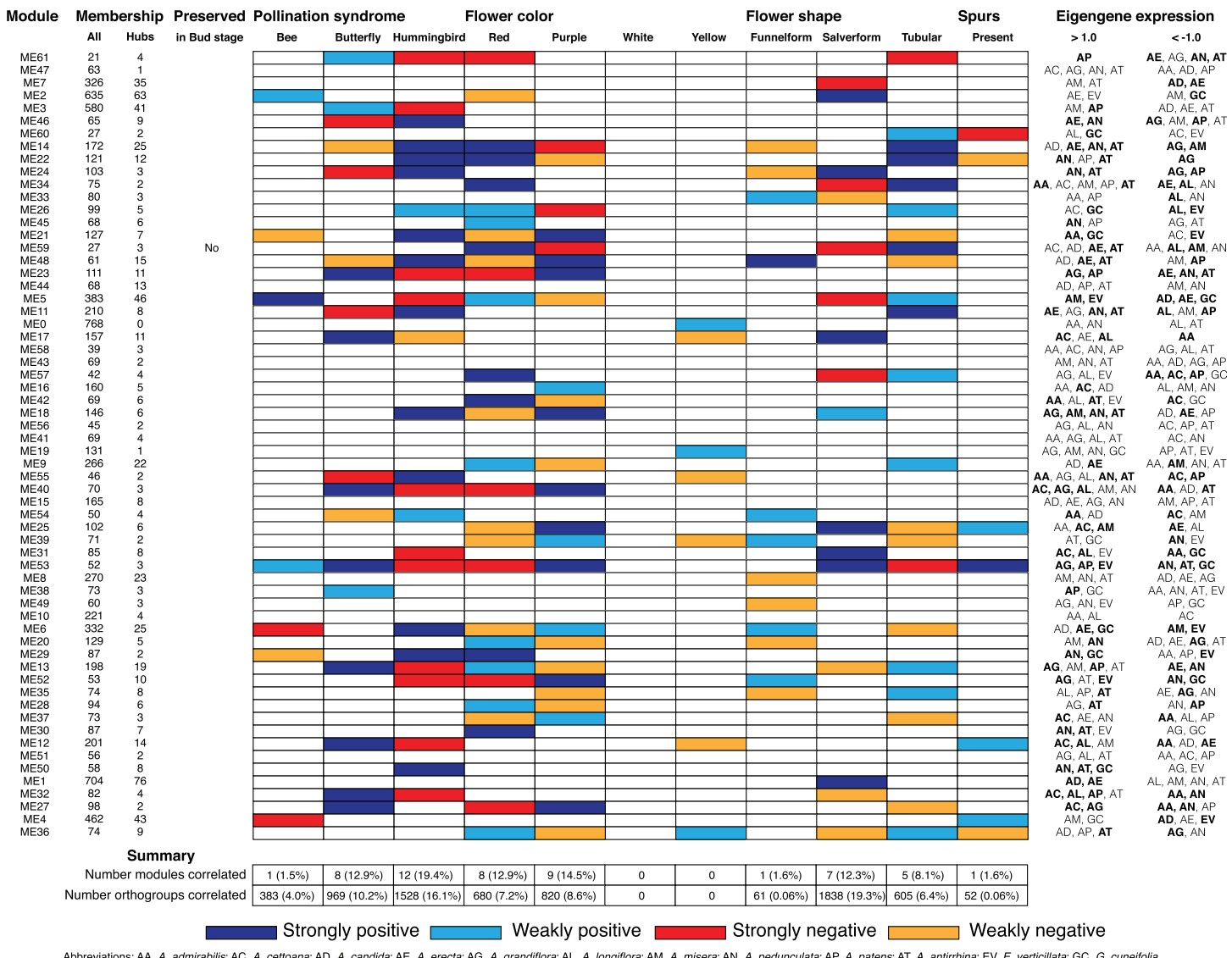

**Figure 6 Summary of D stage network module association to floral traits and pollination syndrome.** Each row represents an individual module in the D stage network and shows the total size (column All), the number of hubs (column Hubs), and whether the module is preserved in the Bud stage. Each column in the central table represents the floral traits (Pollination Syndrome, Flower color, Flower shape, and Spurs) that were tested for an association to module eigengenes. The eigengene expression is indicated for each module in which all three replicates from an individual species had >1.0 or <−1.0 expression. An association was considered strong when both MCMCglmm results were significant and the eigengene expression was >1.0 or <−1.0 in multiple species that shared the significant trait. An association was considered weak when MCMCglmm results were significant but the eigengene expression was >1.0 or <−1.0 in a single species or no species that shared the significant trait. Eigengene expression for species that share traits to those found significant by MCMCglmm are in bold.

elevated eigengene (>1.0) or decreased eigengene (<−1.0) expression in multiple species that shared any of the significant traits. Likewise, weak evidence for trait association was considered when the MCMCglmm results were significant, but only a single species or no species sharing the significant traits had elevated (>1.0) or decreased (<−1.0) eigengene expression. WGCNA does not use trait information when inferring modules, so modules that correlated strongly with any floral trait were inferred without a priori knowledge.

**Table 2 Gene ontology enrichment for the network hubs.**

| GO ID | Term | Annotated | Significant | Expected | p |
|-------|------|-----------|-------------|----------|---|
| (A) Bud stage hubs | | | | | |
| GO:0031325 | Positive regulation of cellular metabolic process | 224 | 16 | 14.52 | 0.000 |
| GO:0006607 | NLS-bearing protein import into nucleus | 7 | 4 | 0.45 | 0.001 |
| GO:0015991 | ATP hydrolysis coupled proton transport | 9 | 4 | 0.58 | 0.002 |
| GO:0046470 | Phosphatidylcholine metabolic process | 9 | 4 | 0.58 | 0.002 |
| GO:0018298 | Protein-chromophore linkage | 32 | 7 | 2.07 | 0.004 |
| GO:0070507 | Regulation of microtubule cytoskeleton organization | 6 | 3 | 0.39 | 0.005 |
| GO:1990778 | Protein localization to cell periphery | 6 | 3 | 0.39 | 0.005 |
| GO:0051592 | Response to calcium ion | 7 | 3 | 0.45 | 0.008 |
| GO:0006412 | Translation | 233 | 24 | 15.10 | 0.009 |
| GO:0042594 | Response to starvation | 60 | 10 | 3.89 | 0.012 |
| GO:0007035 | Vacuolar acidification | 8 | 3 | 0.52 | 0.012 |
| GO:0006534 | Cysteine metabolic process | 9 | 4 | 0.58 | 0.012 |
| GO:0009833 | Plant-type primary cell wall biogenesis | 15 | 4 | 0.97 | 0.013 |
| GO:0048831 | Regulation of shoot system development | 123 | 12 | 7.97 | 0.017 |
| GO:0032509 | Endosome transport via multivesicular body sorting pathway | 9 | 3 | 0.58 | 0.017 |
| GO:0051568 | Histone H3-K4 methylation | 9 | 3 | 0.58 | 0.017 |
| GO:0000271 | Polysaccharide biosynthetic process | 123 | 11 | 7.97 | 0.023 |
| GO:0006732 | Coenzyme metabolic process | 116 | 8 | 7.52 | 0.023 |
| GO:0010338 | Leaf formation | 5 | 2 | 0.32 | 0.037 |
| GO:0048564 | Photosystem I assembly | 5 | 2 | 0.32 | 0.037 |
| GO:0017001 | Antibiotic catabolic process | 26 | 3 | 1.69 | 0.037 |
| GO:0044282 | Small molecule catabolic process | 75 | 10 | 4.86 | 0.038 |
| GO:0006833 | Water transport | 13 | 3 | 0.84 | 0.048 |
| (B) D stage hubs | | | | | |
| GO:0006412 | Translation | 233 | 30 | 16.03 | 0.001 |
| GO:0042939 | Tripeptide transport | 8 | 4 | 0.55 | 0.001 |
| GO:0016117 | Carotenoid biosynthetic process | 20 | 6 | 1.38 | 0.002 |
| GO:0006744 | Ubiquinone biosynthetic process | 5 | 3 | 0.34 | 0.003 |
| GO:0032259 | Methylation | 88 | 15 | 6.05 | 0.004 |
| GO:0045814 | Negative regulation of gene expression, epigenetic | 37 | 5 | 2.55 | 0.005 |
| GO:0009206 | Purine ribonucleoside triphosphate biosynthetic process | 44 | 6 | 3.03 | 0.013 |
| GO:0000271 | Polysaccharide biosynthetic process | 123 | 11 | 8.46 | 0.013 |
| GO:0009873 | Ethylene-activated signaling pathway | 78 | 12 | 5.37 | 0.014 |
| GO:0015991 | ATP hydrolysis coupled proton transport | 9 | 3 | 0.62 | 0.020 |
| GO:0009718 | Anthocyanin-containing compound biosynthetic process | 26 | 5 | 1.79 | 0.030 |
| GO:0000018 | Regulation of DNA recombination | 5 | 2 | 0.34 | 0.041 |
| GO:0015976 | Carbon utilization | 5 | 2 | 0.34 | 0.041 |
| GO:0034314 | Arp2/3 complex-mediated actin nucleation | 5 | 2 | 0.34 | 0.041 |
| GO:0080144 | Amino acid homeostasis | 5 | 2 | 0.34 | 0.041 |
| GO:1901615 | Organic hydroxy compound metabolic process | 121 | 9 | 8.32 | 0.041 |
| GO:0000902 | Cell morphogenesis | 174 | 9 | 11.97 | 0.042 |

**Table 3 Gene ontology enrichment for the network periphery.**

| GO ID | Term | Annotated | Significant | Expected | p |
|-------|------|-----------|-------------|----------|---|
| (A) Bud stage periphery | | | | | |
| GO:0016579 | Protein deubiquitination | 24 | 16 | 8.70 | 0.002 |
| GO:0048825 | Cotyledon development | 21 | 12 | 7.61 | 0.005 |
| GO:0015846 | Polyamine transport | 5 | 5 | 1.81 | 0.006 |
| GO:0034314 | Arp2/3 complex-mediated actin nucleation | 5 | 5 | 1.81 | 0.006 |
| GO:0006020 | Inositol metabolic process | 12 | 7 | 4.35 | 0.011 |
| GO:0010229 | Inflorescence development | 9 | 7 | 3.26 | 0.014 |
| GO:0009744 | Response to sucrose | 31 | 16 | 11.24 | 0.021 |
| GO:1903338 | Regulation of cell wall organization or biogenesis | 11 | 7 | 3.99 | 0.026 |
| GO:0006809 | Nitric oxide biosynthetic process | 6 | 5 | 2.18 | 0.026 |
| GO:0009773 | Photosynthetic electron transport in photosystem I | 8 | 6 | 2.90 | 0.030 |
| GO:0034613 | Cellular protein localization | 183 | 75 | 66.34 | 0.031 |
| GO:0006004 | Fucose metabolic process | 12 | 8 | 4.35 | 0.032 |
| GO:0009926 | Auxin polar transport | 42 | 22 | 15.23 | 0.037 |
| GO:0010090 | Trichome morphogenesis | 38 | 19 | 13.78 | 0.041 |
| GO:0009414 | Response to water deprivation | 182 | 69 | 65.98 | 0.045 |
| GO:0009873 | Ethylene-activated signaling pathway | 78 | 38 | 28.28 | 0.046 |
| GO:0007166 | Cell surface receptor signaling pathway | 93 | 42 | 33.71 | 0.046 |
| GO:0006163 | Purine nucleotide metabolic process | 96 | 38 | 34.80 | 0.047 |
| GO:0035195 | Gene silencing by miRNA | 15 | 7 | 5.44 | 0.048 |
| GO:0016236 | Macroautophagy | 21 | 9 | 7.61 | 0.048 |
| GO:0055083 | Monovalent inorganic anion homeostasis | 13 | 5 | 4.71 | 0.048 |
| GO:0051338 | Regulation of transferase activity | 38 | 13 | 13.78 | 0.048 |
| GO:0006473 | Protein acetylation | 23 | 7 | 8.34 | 0.048 |
| (B) D stage periphery | | | | | |
| GO:0045490 | Pectin catabolic process | 20 | 19 | 12.88 | 0.002 |
| GO:0007166 | Cell surface receptor signaling pathway | 93 | 71 | 59.90 | 0.006 |
| GO:0000398 | mRNA splicing, via spliceosome | 46 | 38 | 29.63 | 0.007 |
| GO:0009740 | Gibberellic acid mediated signaling pathway | 31 | 24 | 19.97 | 0.015 |
| GO:0006310 | DNA recombination | 54 | 41 | 34.78 | 0.015 |
| GO:0009909 | Regulation of flower development | 80 | 57 | 51.53 | 0.017 |
| GO:0009058 | Biosynthetic process | 2,121 | 1,360 | 1,366.19 | 0.019 |
| GO:0042545 | Cell wall modification | 34 | 26 | 21.90 | 0.021 |
| GO:0001708 | Cell fate specification | 13 | 12 | 8.37 | 0.027 |
| GO:0042538 | Hyperosmotic salinity response | 25 | 21 | 16.10 | 0.027 |
| GO:0002181 | Cytoplasmic translation | 21 | 18 | 13.53 | 0.029 |
| GO:0006897 | Endocytosis | 40 | 27 | 25.76 | 0.029 |
| GO:0010197 | Polar nucleus fusion | 20 | 17 | 12.88 | 0.039 |
| GO:0071702 | Organic substance transport | 492 | 319 | 316.91 | 0.046 |
| GO:0006949 | Syncytium formation | 7 | 7 | 4.51 | 0.046 |
| GO:0008295 | Spermidine biosynthetic process | 7 | 7 | 4.51 | 0.046 |
| GO:0009963 | Positive regulation of flavonoid biosynthetic process | 7 | 7 | 4.51 | 0.046 |

### Pollination syndrome

We included three species with the bee pollination syndrome (*A. candida*, *A. misera* and *E. verticillata*), four species with the butterfly pollination syndrome (*A. cettoana*, *A. grandiflora*, *A. longiflora* and *A. patens*), and five species with the hummingbird pollination syndrome (*A. admirabilis*, *A. antirrhina*, *A. erecta*, *A. pedunculata* and *G. cuneifolia*) (Fig. 2; Table 1). These pollination syndromes are traditionally defined in *Achimenes* and other gesneriads based on primary flower color, flower shape, and the presence of corolla spurs (*Wiehler, 1983*; *Ramírez Roa, 1987*; *Roalson, Skog & Zimmer, 2003*). Expression of 23 (35%) and 21 (34%) modules were positively associated with the pollination syndromes in the Bud and D stage networks, consisting of 2,967 and 2,880 orthogroups, respectively (Figs. 5 and 6). There were 3 (4.6%), 7 (10.8%) and 13 (20%) modules associated with bee, butterfly, and hummingbird pollination syndrome in the Bud stage network, while 1 (1.5%), 8 (12.9%) and 12 (19.4%) modules were associated with the same syndromes in the D stage. Modules were never associated with more than one syndrome. Butterfly and hummingbird syndromes were never correlated to the same modules and were often correlated in opposite directions (i.e., butterfly was positive correlated and hummingbird was negative correlated).

Among the three modules that were correlated to the bee pollination syndrome in the Bud stage were many orthogroups involved in hormone signaling pathways (particularly ethylene and abscisic acid, $n = 14$), cell wall and lipid biosynthesis ($n = 7$), photomorphogenesis ($n = 2$), nitrogen compound metabolism ($n = 4$), and the transport of potassium and monocarboxylic acids ($n = 5$) (Table 4). In contrast, the one module correlated to the bee pollination syndrome in the D stage was enriched for different functions from the Bud stage, including the positive regulation of molecular functions and response to biotic stimuli ($n = 7$), phospholipid catalysis ($n = 2$), xyloglucan biosynthesis ($n = 2$), mRNA stability and catalysis ($n = 5$), and chromatin modification ($n = 4$) (Table 5).

The seven modules correlated to the butterfly pollination syndrome in the Bud stage contained orthogroups involved in RNA splicing ($n = 16$), fertilization ($n = 3$), DNA methylation ($n = 2$), cell growth and shoot formation ($n = 8$), the development of organ boundaries ($n = 3$), and stress responses ($n = 83$) (Table 4). Within the eight modules correlated with butterfly pollination in the D stage were orthogroups enriched for functions in photorespiration ($n = 7$), microtubule organization ($n = 5$), chromatin silencing ($n = 3$), red and far-red light responses ($n = 3$), modified amino acid biosynthesis and metabolism ($n = 10$), and mRNA transport ($n = 9$) (Table 5).

Thirteen modules were correlated to the hummingbird pollination syndrome in the Bud stage and the orthogroups were enriched for numerous functions, including many developmental processes (Table 4). These modules were enriched for floral whorl development ($n = 12$), RNA and mRNA modifications ($n = 19$), cell growth and division ($n = 54$), organelle assembly ($n = 3$), transportation of carbohydrates and potassium ($n = 22$), and responses to external stimuli ($n = 17$) (Table 4). Many of the same enrichments were seen in the 12 modules correlated to the hummingbird pollination

**Table 4 Gene ontology enrichment in Bud stage modules correlated to pollination syndrome.**

| GO ID | Term | Annotated | Significant | Expected | p |
|-------|------|-----------|-------------|----------|---|
| (A) Bee pollination | | | | | |
| GO:0046686 | Response to cadmium ion | 99 | 16 | 5.69 | 0.000 |
| GO:0010104 | Regulation of ethylene-activated signaling pathway | 12 | 3 | 0.69 | 0.003 |
| GO:0010189 | Vitamin E biosynthetic process | 8 | 3 | 0.46 | 0.008 |
| GO:0070085 | Glycosylation | 72 | 7 | 4.13 | 0.009 |
| GO:0009226 | Nucleotide-sugar biosynthetic process | 21 | 5 | 1.21 | 0.011 |
| GO:0042136 | Neurotransmitter biosynthetic process | 9 | 4 | 0.52 | 0.018 |
| GO:0070592 | Cell wall polysaccharide biosynthetic process | 32 | 4 | 1.84 | 0.018 |
| GO:0051173 | Positive regulation of nitrogen compound metabolic process | 192 | 4 | 11.03 | 0.019 |
| GO:0046777 | Protein autophosphorylation | 71 | 9 | 4.08 | 0.019 |
| GO:0009651 | Response to salt stress | 215 | 19 | 12.35 | 0.022 |
| GO:0009738 | Abscisic acid-activated signaling pathway | 135 | 11 | 7.75 | 0.028 |
| GO:0015718 | Monocarboxylic acid transport | 12 | 3 | 0.69 | 0.028 |
| GO:0046889 | Positive regulation of lipid biosynthetic process | 12 | 3 | 0.69 | 0.029 |
| GO:0006809 | Nitric oxide biosynthetic process | 5 | 2 | 0.29 | 0.029 |
| GO:0006974 | Cellular response to DNA damage stimulus | 99 | 9 | 5.69 | 0.035 |
| GO:0010099 | Regulation of photomorphogenesis | 6 | 2 | 0.34 | 0.042 |
| GO:0071805 | Potassium ion transmembrane transport | 6 | 2 | 0.34 | 0.042 |
| (B) Butterfly pollination | | | | | |
| GO:0009415 | Response to water | 155 | 13 | 11.47 | 0.016 |
| GO:0008380 | RNA splicing | 122 | 16 | 9.03 | 0.016 |
| GO:0010223 | Secondary shoot formation | 14 | 4 | 1.04 | 0.016 |
| GO:0009624 | Response to nematode | 15 | 4 | 1.11 | 0.021 |
| GO:0009695 | Jasmonic acid biosynthetic process | 9 | 3 | 0.67 | 0.024 |
| GO:0090691 | Formation of plant organ boundary | 9 | 3 | 0.67 | 0.024 |
| GO:0006950 | Response to stress | 1,173 | 83 | 86.79 | 0.028 |
| GO:0006353 | DNA-templated transcription, termination | 17 | 4 | 1.26 | 0.032 |
| GO:0009567 | Double fertilization forming a zygote and endosperm | 11 | 3 | 0.81 | 0.042 |
| GO:0009835 | Fruit ripening | 11 | 3 | 0.81 | 0.042 |
| GO:0009607 | Response to biotic stimulus | 374 | 35 | 27.67 | 0.045 |
| GO:0010216 | Maintenance of DNA methylation | 5 | 2 | 0.37 | 0.047 |
| GO:0010363 | Regulation of plant-type hypersensitive response | 5 | 2 | 0.37 | 0.047 |
| GO:0071577 | Zinc ion transmembrane transport | 5 | 2 | 0.37 | 0.047 |
| GO:0009825 | Multidimensional cell growth | 19 | 4 | 1.41 | 0.047 |
| (C) Hummingbird pollination | | | | | |
| GO:0009451 | RNA modification | 353 | 89 | 62.34 | 0.000 |
| GO:1900618 | Regulation of shoot system morphogenesis | 9 | 5 | 1.59 | 0.001 |
| GO:0009826 | Unidimensional cell growth | 95 | 24 | 16.78 | 0.002 |
| GO:0016556 | mRNA modification | 35 | 11 | 6.18 | 0.002 |
| GO:0006012 | Galactose metabolic process | 5 | 4 | 0.88 | 0.004 |
| GO:0031425 | Chloroplast RNA processing | 18 | 8 | 3.18 | 0.008 |

(Continued)

| GO ID | Term | Annotated | Significant | Expected | p |
|---|---|---|---|---|---|
| GO:0008643 | Carbohydrate transport | 44 | 15 | 7.77 | 0.010 |
| GO:1905428 | Regulation of plant organ formation | 6 | 4 | 1.06 | 0.011 |
| GO:0006888 | ER to Golgi vesicle-mediated transport | 19 | 8 | 3.36 | 0.011 |
| GO:0044272 | Sulfur compound biosynthetic process | 58 | 12 | 10.24 | 0.011 |
| GO:0045944 | Positive regulation of transcription by RNA polymerase II | 37 | 13 | 6.53 | 0.013 |
| GO:0002239 | Response to oomycetes | 18 | 6 | 3.18 | 0.019 |
| GO:0006813 | Potassium ion transport | 16 | 7 | 2.83 | 0.020 |
| GO:0043650 | Dicarboxylic acid biosynthetic process | 10 | 5 | 1.77 | 0.020 |
| GO:0051053 | Negative regulation of DNA metabolic process | 7 | 4 | 1.24 | 0.022 |
| GO:0051301 | Cell division | 147 | 30 | 25.96 | 0.027 |
| GO:0000079 | Regulation of cyclin-dependent protein serine/threonine kinase activity | 11 | 5 | 1.94 | 0.031 |
| GO:0046131 | Pyrimidine ribonucleoside metabolic process | 12 | 3 | 2.12 | 0.031 |
| GO:0032509 | Endosome transport via multivesicular body sorting pathway | 8 | 4 | 1.41 | 0.037 |
| GO:0071496 | Cellular response to external stimulus | 71 | 17 | 12.54 | 0.041 |
| GO:1902117 | Positive regulation of organelle assembly | 5 | 3 | 0.88 | 0.041 |
| GO:0048438 | Floral whorl development | 69 | 12 | 12.19 | 0.042 |
| GO:0009651 | Response to salt stress | 215 | 49 | 37.97 | 0.043 |

**Table 5 Gene ontology enrichment in D stage modules correlated to pollination syndrome.**

| GO ID | Term | Annotated | Significant | Expected | p |
|---|---|---|---|---|---|
| (A) Bee pollination | | | | | |
| GO:0061014 | Positive regulation of mRNA catabolic process | 6 | 3 | 0.26 | 0.002 |
| GO:0010189 | Vitamin E biosynthetic process | 8 | 3 | 0.35 | 0.004 |
| GO:0044093 | Positive regulation of molecular function | 46 | 4 | 2.01 | 0.006 |
| GO:0046686 | Response to cadmium ion | 99 | 10 | 4.33 | 0.011 |
| GO:0006974 | Cellular response to DNA damage stimulus | 99 | 8 | 4.33 | 0.017 |
| GO:0002833 | Positive regulation of response to biotic stimulus | 19 | 3 | 0.83 | 0.017 |
| GO:0009395 | Phospholipid catabolic process | 5 | 2 | 0.22 | 0.017 |
| GO:0009969 | Xyloglucan biosynthetic process | 5 | 2 | 0.22 | 0.017 |
| GO:0006544 | Glycine metabolic process | 7 | 2 | 0.31 | 0.035 |
| GO:0043488 | Regulation of mRNA stability | 7 | 2 | 0.31 | 0.035 |
| GO:0016569 | Covalent chromatin modification | 58 | 4 | 2.54 | 0.044 |
| GO:0043902 | Positive regulation of multi-organism process | 22 | 3 | 0.96 | 0.045 |
| GO:0009631 | Cold acclimation | 8 | 2 | 0.35 | 0.045 |
| (B) Butterfly pollination | | | | | |
| GO:0009853 | Photorespiration | 19 | 7 | 2.00 | 0.002 |
| GO:0043622 | Cortical microtubule organization | 13 | 5 | 1.37 | 0.008 |
| GO:0031349 | Positive regulation of defense response | 51 | 8 | 5.37 | 0.010 |
| GO:0006575 | Cellular modified amino acid metabolic process | 35 | 7 | 3.69 | 0.018 |

| Table 5 (continued). | | | | | |
|---|---|---|---|---|---|
| **GO ID** | **Term** | **Annotated** | **Significant** | **Expected** | **p** |
| GO:0070919 | Production of siRNA involved in chromatin silencing by small RNA | 6 | 3 | 0.63 | 0.018 |
| GO:0042398 | Cellular modified amino acid biosynthetic process | 7 | 3 | 0.74 | 0.029 |
| GO:0009808 | Lignin metabolic process | 22 | 5 | 2.32 | 0.031 |
| GO:2000030 | Regulation of response to red or far red light | 9 | 3 | 0.95 | 0.031 |
| GO:0019220 | Regulation of phosphate metabolic process | 39 | 6 | 4.11 | 0.031 |
| GO:0010197 | Polar nucleus fusion | 13 | 4 | 1.37 | 0.040 |
| GO:0051028 | mRNA transport | 44 | 9 | 4.63 | 0.041 |
| GO:0006833 | Water transport | 13 | 4 | 1.37 | 0.043 |
| GO:0043902 | Positive regulation of multi-organism process | 22 | 4 | 2.32 | 0.044 |
| (C) Hummingbird pollination | | | | | |
| GO:0009960 | Endosperm development | 10 | 6 | 1.53 | 0.002 |
| GO:0007163 | Establishment or maintenance of cell polarity | 5 | 4 | 0.77 | 0.002 |
| GO:0000272 | Polysaccharide catabolic process | 50 | 14 | 7.65 | 0.005 |
| GO:0010224 | Response to UV-B | 24 | 9 | 3.67 | 0.007 |
| GO:0009089 | Lysine biosynthetic process via diaminopimelate | 7 | 4 | 1.07 | 0.013 |
| GO:0009451 | RNA modification | 353 | 70 | 54.01 | 0.015 |
| GO:0007166 | Cell surface receptor signaling pathway | 86 | 21 | 13.16 | 0.018 |
| GO:0008285 | Negative regulation of cell proliferation | 8 | 4 | 1.22 | 0.023 |
| GO:0042939 | Tripeptide transport | 8 | 4 | 1.22 | 0.023 |
| GO:0009955 | Adaxial/abaxial pattern specification | 17 | 8 | 2.60 | 0.023 |
| GO:0051656 | Establishment of organelle localization | 36 | 9 | 5.51 | 0.023 |
| GO:0006213 | Pyrimidine nucleoside metabolic process | 14 | 3 | 2.14 | 0.023 |
| GO:0033014 | Tetrapyrrole biosynthetic process | 39 | 6 | 5.97 | 0.024 |
| GO:0009267 | Cellular response to starvation | 47 | 14 | 7.19 | 0.025 |
| GO:0006635 | Fatty acid beta-oxidation | 16 | 6 | 2.45 | 0.026 |
| GO:0016556 | mRNA modification | 35 | 9 | 5.36 | 0.026 |
| GO:0048438 | Floral whorl development | 69 | 14 | 10.56 | 0.028 |
| GO:0010103 | Stomatal complex morphogenesis | 14 | 5 | 2.14 | 0.028 |
| GO:0006383 | Transcription by RNA polymerase III | 5 | 3 | 0.77 | 0.028 |
| GO:0040008 | Regulation of growth | 126 | 26 | 19.28 | 0.029 |
| GO:0010019 | Chloroplast-nucleus signaling pathway | 9 | 4 | 1.38 | 0.036 |
| GO:0010506 | Regulation of autophagy | 9 | 4 | 1.38 | 0.036 |
| GO:0043547 | Positive regulation of GTPase activity | 9 | 4 | 1.38 | 0.036 |
| GO:0018298 | Protein-chromophore linkage | 27 | 8 | 4.13 | 0.044 |
| GO:0031425 | Chloroplast RNA processing | 18 | 6 | 2.75 | 0.045 |
| GO:0051187 | Cofactor catabolic process | 34 | 9 | 5.20 | 0.050 |
| GO:0002238 | Response to molecule of fungal origin | 6 | 3 | 0.92 | 0.050 |
| GO:0032958 | Inositol phosphate biosynthetic process | 6 | 3 | 0.92 | 0.050 |
| GO:0042219 | Cellular modified amino acid catabolic process | 6 | 3 | 0.92 | 0.050 |
| GO:1905428 | Regulation of plant organ formation | 6 | 3 | 0.92 | 0.050 |
| GO:0065004 | Protein-DNA complex assembly | 45 | 8 | 6.89 | 0.050 |

syndrome in the D stage. These modules had enrichments for floral whorl development ($n = 14$), cell polarity and proliferation ($n = 8$), adaxial/abaxial specification ($n = 8$), organ formation ($n = 3$), the regulation of growth ($n = 26$), and endosperm development ($n = 6$). Additionally, these modules had enrichment for RNA and mRNA modification ($n = 79$), transcription ($n = 3$), protein-DNA complex formation ($n = 7$), signaling ($n = 25$), and UV response ($n = 9$) (Table 5).

### Flower color

Primary flower color is closely associated to pollination syndrome in *Achimenes* (*Ramírez Roa, 1987*; *Roalson, Skog & Zimmer, 2003*), and we identified 26 (40%) and 17 (27%) modules associated to any flower color in the Bud and D stage networks, consisting of 2,935 and 1,500 orthogroups respectively (Figs. 5 and 6). Red and yellow are associated with hummingbird pollination, purple with butterfly pollination, and white with bee pollination. In the Bud stage, there were 14 (21.5%), 11 (16.9%) and 1 (1.5%) module associated with red, purple, or yellow flower color. In the D stage, no modules were associated with yellow, but 8 (12.9%) and 9 (14.5%) modules were associated with red or purple flower color. In both networks, no modules were corelated to white flower color. Similar to syndrome, purple and red flower color were never associated with the same modules and were always correlated in opposite directions.

The 14 modules associated with red flowers in the Bud stage overlapped significantly with the modules associated with the hummingbird pollination syndrome in the Bud stage and contained the same functional enrichments (Table 4; Table S8). However, in the 8 D stage modules associated with red flowers, the orthogroups were enriched for functions involved in flower morphogenesis ($n = 8$), cell wall modification ($n = 3$), positive regulation of gene expression ($n = 14$), and histone modification ($n = 7$), among others (Table S8).

Within the 11 modules associated with purple flowers in the Bud stage were orthogroups enriched for transcription ($n = 14$), meristem initiation ($n = 4$), DNA methylation ($n = 3$), fertilization ($n = 5$), ethylene signaling ($n = 5$), response to biotic stimuli ($n = 55$), and others (Table S8). In contrast, the nine modules associated with purple flowers in the D stage were enriched for orthogroups involved in anther and endosperm development ($n = 9$), translation ($n = 4$), the regulation of developmental processes ($n = 9$), lipid modification ($n = 8$), cell division ($n = 3$), and hormone signaling (gibberellic acid and jasmonic acid, $n = 6$), among many other processes (Table S8).

A single module (ME36) was associated with yellow flower color, only in the Bud stage, and contained few orthogroups involved in floral meristem determinacy ($n = 1$), floral organ identity ($n = 1$), gene silencing ($n = 1$), sulfur compound biosynthesis ($n = 2$), and vegetative phase change ($n = 1$) (Table S8).

### Flower shape

Flower shape is also important but not as closely associated to pollination syndromes as primary flower color in *Achimenes* (*Ramírez Roa, 1987*; *Roalson, Skog & Zimmer, 2003*). Flowers with bee pollination tend to have funnelform flowers, butterfly pollinated flowers all have salverform flowers, and hummingbird pollinated flowers have either

salverform or tubular flowers (Table 2). Overall, there were 14 (22%) and 13 (21%) modules associated with any of the flower shapes in the Bud and D stages, consisting of 2,940 and 2,504 orthogroups, respectively (Figs. 5 and 6). Here, there were 1 (1.5%), 8 (12.3%) and 5 (7.7%) modules whose expression correlated to funnelform, salverform, or tubular flowers in the Bud stage (Fig. 5). Fewer modules were correlated in the D stage network, but expression in 1 (1.6%), 7 (12.3%) and 5 (8.1%) modules were associated with funnelform, salverform, and tubular shapes (Fig. 4). No modules were positively associated with more than one flower shape.

Only two modules were associated with funnelform flowers, one in the Bud stage and the other in the D stage, and these were enriched for functions related to ethylene signaling ($n = 3$), lipid biosynthesis ($n = 3$), chromatin modification ($n = 4$), cell wall organization ($n = 2$), and many metabolic processes (Table S8). Within the eight modules associated with salverform flowers in the Bud stage, there were many orthogroups involved in mRNA processing and modification ($n = 50$), ovule development ($n = 7$), chlorophyll biosynthesis ($n = 4$), and the regulation of flower development ($n = 20$) (Table S8). In the D stage, the seven modules associated with salverform flowers contained orthogroups involved in chromatin silencing ($n = 7$), anthocyanin biosynthesis ($n = 10$), cell cycle and cell division ($n = 95$), stamen development ($n = 12$), auxin biosynthesis ($n = 7$), and transport ($n = 17$) (Table S8).

Among the five modules associated with tubular flowers in the Bud stage were orthogroups enriched for functions in the positive regulation of transcription and organelle assembly ($n = 13$), RNA processing ($n = 5$), xylem development ($n = 4$), metabolism ($n = 12$), and many biosynthetic processes (Table S8). In the five modules associated with tubular flowers in the D stage were enrichments for floral whorl development ($n = 6$), positive regulation of gene expression ($n = 15$), mRNA modification ($n = 5$), and signaling ($n = 12$) (Table S8).

### Corolla spurs

The presence of corolla spurs is found only on butterfly pollinated species, two of which were included: *A. grandiflora* and *A. patens* (Fig. 1; Table 1). There was expression in 5 (7.7%) modules associated with spurs in the Bud stage network, while only 1 (1.6%) module was associated in the D stage network (Figs. 3 and 4). Within the five modules associated with corolla spurs in the Bud stage were orthogroups related to miRNA processing ($n = 4$), mRNA splicing ($n = 9$), the regulation of signal transduction pathways ($n = 16$), and pollen tube growth ($n = 6$) (Table S8). In the single module associated with corolla spurs in the D stage (ME53) were very few orthogroups involved in microtubule organization ($n = 2$), stamen development ($n = 1$), the positive regulation of cell division ($n = 1$), flavonoid biosynthesis ($n = 1$), and indole acetic acid metabolism ($n = 1$) (Table S8).

### Non-associated

There were 13 and nine modules not associated with any traits in the Bud and D stage networks, respectively (Figs. 5 and 6). Three of these modules in the Bud stage were

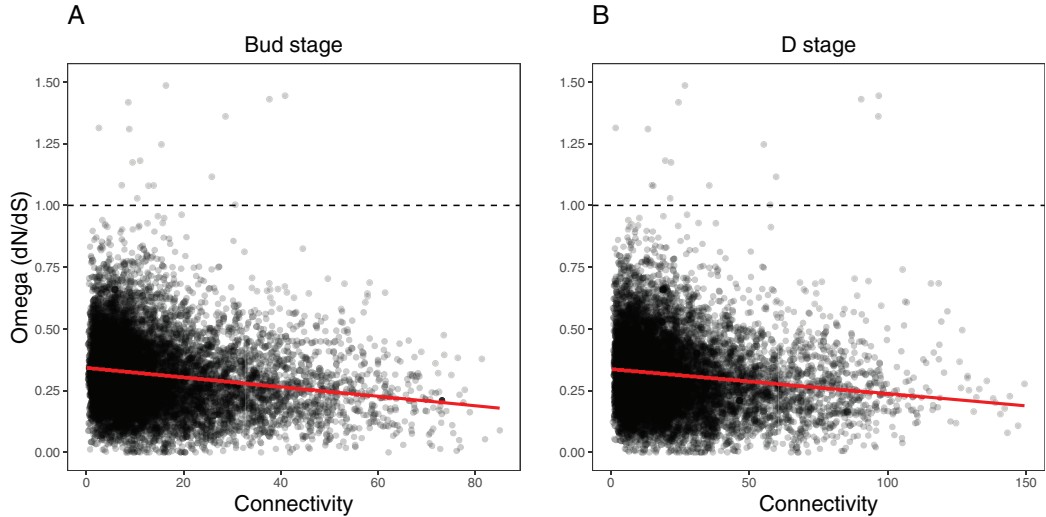

**Figure 7 Relationship between orthogroup $d_N/d_S$ (omega) and orthogroup connectivity.** (A) Bud stage network. (B) D stage network. Each point represents an individual orthogroup. The red line indicates the linear regression line for the relationship between $d_N/d_S$ and connectivity. The dotted line indicates $d_N/d_S = 1$ with orthogroups above this line considered to be under relaxed or positive selection.

not-preserved in the D stage (modules ME46, ME44, and ME41; Figs. 5 and 6; Table S5). The nine non-associated modules in the D stage were also not associated with any traits in the Bud stage (Figs. 5 and 6) and the orthogroups had overlapping enrichments for many essential processes (Table S8). These modules were enriched for orthogroups involved in the development of the inflorescence, stamens, and floral organs, biosynthesis of vitamins and nucleotides, photosynthesis and photosynthetic electron transport, metabolism of phosphate compounds, and cell surface receptor signaling (Table S8). Numerous orthogroups were also involved in the positive regulation of reproduction, the negative regulation of growth, and the regulation of heterochronic development, histone methylation, and gibberellic acid mediated signaling (Table S8).

## Evolutionary rates correlate to network location

We ran several analyses to explore how several network variables may affect evolutionary rates of the orthogroups within each network (as measured by $d_N/d_S$). First, we tested the effects of orthogroup connectivity, expression levels, and their interaction on the $d_N/d_S$ values (Table S7). Results indicated a collective effect between connectivity and expression on $d_N/d_S$ in both the Bud stage (LM, $F_{(3, 9,548)} = 92.58$, $p = 0$, $\eta^2 = 0.03$) and the D stage networks (LM, $F_{(3, 9,548)} = 78.36$, $p = 0$, $\eta^2 = 0.02$). We found that orthogroup connectivity was negatively correlated with $d_N/d_S$ in the Bud stage network (LM, $t = -16.31$, $p = 0$) and the D stage network (LM, $t = -15.13$, $p = 0$) (Fig. 7), while expression was positively correlated with $d_N/d_S$ in the Bud stage (LM, $t = 2.12$, $p = 0.0344$) and not in the D stage (LM, $t = 1.51$, $p = 0.1324$) (Fig. S7). The interaction between connectivity and expression also indicated a marginal dependency of both predictors on $d_N/d_S$ in the Bud stage

(LM, $t = -2.022$, $p = 0.0433$) but not in the D stage network (LM, $t = -1.823$, $p = 0.0683$). These patterns have been observed in model species and may be universal to all eukaryotes.

Second, we tested whether the nodes we defined as hubs in each network had lower $d_N/d_S$ values, suggestive of increased evolutionary constraint (Table S9). Hubs are considered to functionally significant and may control large portions of the network. We found that the hubs in the Bud stage network had lower $d_N/d_S$ than all background genes (Two-sample $t$-test, $t_{(679.31)} = 7.0934$, $p = 1.65e-12$, Cohen's $d = 0.383$; Permutation $t$-test. mean diff. $= 0.266$, $p = 0.0001$) (Fig. S8). Similarly, hubs in the D stage network also had lower $d_N/d_S$ than all background genes (Two-sample $t$-test, $t_{(755.91)} = 4.1918$, $p = 1.55e-05$, Cohen's $d = 0.206$; Permutation $t$-test, mean diff. $= 0.143$, $p = 0.0001$) (Fig. S8).

Third, we tested whether the most peripheral nodes (the lowest 10% connected nodes) had higher $d_N/d_S$ values, which may suggest relaxed evolutionary constraint (Table S9). These nodes are loosely connected within the network and their roles may be more apt to fluctuate during evolution and development. These peripheral genes had higher $d_N/d_S$ than all background genes in the Bud stage network (Two-sample $t$-test, $t_{(1,487.70)} = -10.272$, $p = 0$, Cohen's $d = -0.253$; Permutation $t$-test, mean diff. $= -0.176$, $p = 0.0001$) and the D stage network (Two-sample, $t_{(1,512.30)} = -10.387$, $p = 0$, Cohen's $d = -0.252$; Permutation $t$-test, mean diff. $= -0.175$, $p = 0.0001$) (Fig. S8).

Lastly, we tested whether modules correlated with bee, butterfly, or hummingbird pollination syndromes showed increased evolutionary rates over the non-associated modules. Results indicated a very weak effect of syndrome association on $d_N/d_S$ in the Bud network (LM, $F_{(3,\ 9,548)} = 5.186$, $p = 0.001406$, $\eta^2 = 0.0016$) and the D network (LM, $F_{(3,\ 9,548)} = 4.277$, $p = 0.005038$, $\eta^2 = 0.0013$) (Table S9). Only bee syndrome associated modules had marginally lower $d_N/d_S$ values in both networks (Bud stage, LM, $t = -3.58$, $p = 0.000346$; D stage, LM, $t = -2.508$, $p = 0.0122$). Butterfly associated modules had marginally lower $d_N/d_S$ only in the D stage (LM, $t = -2.548$, $p = 0.0108$), while hummingbird associated modules did not have any effect in either network.

## DISCUSSION

Here we report on patterns of gene co-expression across two stages of flower development in 12 species of *Achimenes*, *Eucodonia* and *Gesneria*. Our comparative and phylogenetic context allowed us to reveal gene co-expression patterns across a group of closely related plant species that correlated to diverse pollination syndromes and floral traits, including flower color, flower shape, and corolla spurs. Our results suggest floral forms that are associated with bee, butterfly, and hummingbird pollinator groups are correlated with many co-expression modules. We found that nearly a third of modules in each network (23 and 21 modules in the Bud and D stages, respectively) had evidence for association with the three pollination syndromes (Figs. 5 and 6). In our analysis, large fractions of our 9,503 orthogroups set could be partitioned into modules (Fig. 3), network hubs and peripheral nodes identified (Tables S6 and S7), and several orthogroups were identified for their potential importance in floral phenotypic differentiation. Lastly, orthogroup position in each network (measured by connectivity) had an association with evolutionary rate

($d_N/d_S$) (Fig. 7). These analyses provide a first overview of floral transcriptional architecture across members of the same genus that display diverse floral forms.

## Hub nodes contain important candidates for floral form

To date, a large number of studies conducted in diverse angiosperm systems, have found that many genes are associated with flower development (*Quattrocchio et al., 1999*; *Howarth et al., 2011*). Despite evidence for extensive gene/genome duplications that could expand the genetic repertoire (*Tang et al., 2008*), many recent studies have suggested that the generation of novel genes is rare (*Paterson et al., 2009*; *De Smet et al., 2013*). Thus, the evolution and development of diverse floral forms may predominantly occur through the co-option of existing pathways (*Preston, Hileman & Cubas, 2011*). To our knowledge, gene co-expression network analyses have not been previously used across multiple species in a non-model lineage to understand flower development and evolution.

Hub nodes (genes) are considered to be conserved, highly connected nodes that are central to network architecture and may directly orchestrate numerous biological processes (*Ravasz et al., 2002*; *Hu et al., 2016*; *Mähler et al., 2017*). Our analyses identified over 600 gene nodes as hubs within each network, the majority of which were only hubs within a specific floral stage. These hub nodes tended to have lower $d_N/d_S$ estimates, suggesting that they not prone to rapid evolutionary change (Table S9). Unsurprisingly, hub nodes in the Bud stage were enriched for many processes related to early flower development, including cell growth, leaf formation, and photosystem assembly (Table 2). Similarly, hub nodes in the D stage were enriched for genes that may be related the development of distinct floral forms, including carotenoid biosynthesis, anthocyanin biosynthesis, and cell morphogenesis (Table 2). Within this context of flower development, we identified a number of hubs that we consider important candidates for involvement in the production of flower color and the development of floral form. Many of these candidates were considered previously (*Roberts & Roalson, 2017*) as potentially important for the diversification of floral form in *Achimenes*. Undoubtedly, this list of hubs contains many genes that may important for other biological traits that we do not consider here, such as scent, nectar, and pollen production.

### *Anthocyanin biosynthesis*

Anthocyanins are common floral pigments that produce red, purple, and blue colors and are produced via the highly conserved flavonoid biosynthetic pathway (*Holton & Cornish, 1995*). With a few exceptions, all anthocyanins are produced from three precursor molecules: pelargonidin, cyanidin, and delphinidin (*Grotewold, 2006*). There are six enzymatic reactions required for the production of anthocyanins and many also serve as branching points for the production of flavonoids, lignins, flavonols, and other phenolics that have roles in stress response and physiology (*Winkel-Shirley, 2002*; *Tahara, 2007*). We identified four homologs of anthocyanin biosynthetic pathway enzymes as network hubs: flavonoid 3′-hydroxylase (*F3′H*; cluster8703_1rr.inclade1.ortho3; module ME12), dihydroflavonol 4-reductase (*DFR*; cluster5082_1rr.inclade1.ortho6; module ME48), flavonoid 3′,5′-methyltransferase (*FAOMT*; cluster11783_1rr.inclade1.ortho1.tre; module

ME32), and flavonol 3-*O*-glucosyltransferase (*UFOG*; cluster8241_2rr.include1.ortho2.tre; module ME1).

These four enzymes are involved in distinct steps during anthocyanin production and chemical modification in floral tissue. Activity of *F3'H* directs the pathway toward the production of cyanidins and quercetin (*Grotewold, 2006*). Cyanidin can accumulate in floral tissue as a purple-hued pigment, while quercetin plays a large role in various stress-related physiological functions (*Pollastri & Tattini, 2011*), including response to high light and UVB radiation (*Gerhardt, Lampi & Greenberg, 2008*) and antioxidant properties (*Rice-Evans, Miller & Papanga, 1996*). Therefore, *F3'H* as an important hub in the networks makes sense given its enormous pleiotropic role in various physiological responses. Just prior to the production of anthocyanins, *DFR* catalyzes the reaction to produce pelargonidins, cyanidins, and delphinidins from the precursor molecules dihydrokaempferol, dihydroquercetin, and dihyrdomyricetin, respectively (*Grotewold, 2006*). Shifts in *DFR* expression, function, and substrate specificity have been implicated in many evolutionary transitions in flower color (*Smith, Wang & Rausher, 2013*; *Wu et al., 2013*).

Further modifications to anthocyanins, such as methylation, glycosylation, and acylation, produce a wide variety of compounds and hues (*Wessinger & Rausher, 2012*). We identified homologs to both a methyltransferase (*FAOMT*) and glucosyltransferase (*UFOG*) as hubs in the networks. Methylation of anthocyanins have been widely studied in petunia (*Provenzano et al., 2014*), grapes (*Hugueney et al., 2009*) and tree peony (*Du et al., 2015*), demonstrating that methylation can impact the hue, either shifting it more toward purple (*Sakata et al., 1995*) or sometimes red (*Tanaka, Sasaki & Ohmiya, 2008*). Glycosylation stabilizes anthocyanins molecules for subsequent transfer to acidic vacuoles (*Saito & Yamazaki, 2002*), increasing their solubility, and influencing their color variability by inter- and intra-molecular stacking (*Springob et al., 2003*). Finding homologs of *FAOMT* and *UFOG* as network hubs reflect the importance of anthocyanin modification for different flower color hues among *Achimenes* species.

### Carotenoid biosynthesis

Carotenoids are red, orange, and yellow pigments that are produced in plastids and are essential for photosynthesis, but also accumulate as secondary metabolites in flowers to attract pollinators (*Kevan & Baker, 1983*). Floral and fruit carotenoids can also be used to produce volatile apocarotenoids, which may further enhance plant–animal interactions during pollination and seed dispersal (*Dudareva et al., 2006*). They may also serve as precursors to the production of the hormone abscisic acid and strigalactones that are involved in many plant developmental processes (*Rosas-Saavedra & Stange, 2016*). We identified four homologs of carotenoid biosynthetic pathway enzymes as network hubs: geranylgeranyl pyrophosphate synthase (*GGPPS*; cluster10945_2rr.include1.ortho1.tre; module ME35), phytoene dehydrogenase (*PDS*; cluster13048_1rr.include1.ortho1.tre; module ME4), lycopene beta cyclase (*LCYB*; cluster13774_1rr.include1.ortho1.tre; module ME7), and capsanthin-capsorubin synthase (*CCS*; cluster13583_1rr.include1.ortho2.tre; module ME6).

Each of these four enzymes catalyzes reactions during different steps of carotenoid production. *GGPPS* produces the main precursor molecular geranylgeranyl pyrophosphate (GGPP) to carotenoid biosynthesis that is produced as the end-product of the plastid methylerythritol 4-phosphate (MEP) pathway. Two GGPP molecules are then condensed into phytoene and undergoes four successive dehydrogenations and isomerizations to produce lycopene. *PDS* catalyzes the first two desaturation steps that transform the colorless phytoene into the red-colored lycopene. *PDS* is a rate-limiting enzyme in carotenoid biosynthesis (*Chamovitz, Sandmann & Hirschberg, 1993*) and down-regulation has been shown to lead to large accumulations of phytoene (*Busch, Seuter & Hain, 2002*) and decreases in total carotenoid, chlorophyll, and photosynthetic efficiency (*Wang et al., 2009*).

Lycopene is the main starting compound for a large variety of carotenoids. *LCYB* takes lycopene as a substrate and subsequently produces beta-carotene. Many plants use tissue-specific isoforms of *LCYB* genes that are expressed in fruits or flowers that often correlate with the accumulation of beta-carotene or downstream xanthophylls (*Ronen et al., 2000*; *Ahrazem et al., 2010*; *Devitt et al., 2010*). Beta-carotenes are hypothesized to be the primary carotenoids in *Achimenes* flowers (*Roberts & Roalson, 2017*). Another functionally related carotenoid cyclase enzyme is *CCS* (or neoxanthin synthase, *NSY*) which catalyzes the final step in the carotenoid pathway and converts the yellow-colored violaxanthin into neoxanthin (*Parry & Horgan, 1991*). Both violaxanthin and neoxanthin can be further used for producing the plant hormone abscisic acid (*Neuman et al., 2014*). Both enzymes are likely functionally important in the developing flowers for the biosynthesis of beta-carotene and xanthophylls.

### Flower development

Very few studies have so far investigated the molecular mechanisms involved in the development of flower shape in the Gesneriaceae (*Gao et al., 2008*; *Zhou et al., 2008*; *Alexandre et al., 2015*). However, extensive work has been done in model organisms, such as *Arabidopsis thaliana* and *Antirrhinum majus*, which can provide insight into the potential roles that many transcription factors might play in determining floral organ identity and shape in *Achimenes*. Among the network hubs, we identified several homologs that are likely involved in floral organ development, including mixta (*MIXTA*; cluster8699_1rr.inclade1.ortho3.tre; modules ME5 and ME8), agamous (*AG*; cluster7311_1rr.inclade1.ortho3.tre; module ME23), globosa (*GLO*; cluster2750_1rr.inclade1.ortho3.tre; module ME37), and tcp4 (*TCP4*; cluster12380_1rr.inclade1.ortho2.tre; module ME22).

Petal identity and petal surface morphology are both thought to be important components for pollinator attraction (*Noda et al., 1994*; *Whitney et al., 2011*). *MIXTA* was first characterized from *Antirrhinum* and controls the development of the conical cell shape in the petal epidermis (*Noda et al., 1994*). Consequently, *MIXTA* and *MIXTA*-like genes have been characterized in other model systems, such as *Arabidopsis* and *Mimulus* for their similar role in epidermal cell morphogenesis (*Baumann et al., 2007*) as well as trichome development (*Gilding & Marks, 2010*; *Scoville et al., 2011*). The homolog of

*MIXTA* in our study was a hub node in both networks, suggesting it might play an important role throughout flower development in determining petal surface morphology.

Early floral organ determination requires the involvement of several MADS-box transcription factors, such as *AG* and *GLO*. We identified homologs of *AG* as a hub in the Bud stage and *GLO* as a hub in the D stage (Table S7). *AG* has been shown in *Arabidopsis* to be expressed in early flower development in order to terminate meristem activity and promote the development of stamens and carpels, while working in combination with *APETALA3*, *PISTILLATA* and *SEPALLATA* (*Gómez-Mena et al., 2005*). More recently homologs of *AG* were demonstrated to be involved in nectary development in both *Arabidopsis* and *Petunia* (*Morel et al., 2018*), another floral trait important for plant-pollinator interaction. *GLO* was first characterized in *Antirrhinum* and *Arabidopsis* and is also involved with stamen identity, as well as petal identity (*Tröbner et al., 1992*; *Goto & Meyerowitz, 1994*). Outside of these model systems, its expression patterns and role in the evolution of complex floral forms have been examined in numerous plant lineages, from monocots (*Bartlett & Specht, 2010*) to Solanaceae (*Zhang et al., 2014*). *AG* and *GLO* will be interesting candidates to begin exploring further how petals, stamens, carpels and nectary development in *Achimenes*.

Lastly, the homolog of *TCP4* was identified as a hub during the Bud stage, and may be important for its potential role in petal development and cell proliferation. Looking into the genetic factors regulating the elaboration of petal growth will be important for understanding how corolla spurs develop in *Achimenes* (Fig. 1). The role of *TCP4* been characterized in *Arabidopsis* for its role in petal growth, showing restricted expression to the developing petal tissue (*Nag, King & Jack, 2009*). More recent work using gene regulatory networks in *Arabidopsis* has shown that petal size is controlled by a *SEPALLATA3*-regulated miR319/*TCP4* module (*Chen et al., 2018*). Outside of *Arabidopsis*, *TCP4* was also implicated for a having a role in the development of the petal spur in *Aquilegia* (*Yant et al., 2015*). Therefore, considering both the role for *TCP4* in petal and spur development demonstrated in *Arabidopsis* and *Aquilegia* and its identification as a hub gene in our networks, we think this gene requires further interrogation.

## Peripheral genes are enriched for transcriptional regulators in later stages of development

Peripheral nodes have low connectivity within co-expression networks (Table S7), tend to have higher rates of molecular evolution (*Masalia, Bewick & Burke, 2017*), and have been hypothesized to contribute more to evolutionary innovations than network hubs (*Ichihashi et al., 2014*). Of the 951 nodes we defined as peripheral in both Bud and D stage networks, roughly a third were consistently peripheral under our definitions (lowest 10% connected; Table S7). After identifying the nodes with homologies to transcriptional regulators, we found there was no difference in the expected number in the early Bud stage, while there were greater than expected numbers of transcriptional regulators in the periphery of the D stage (see "Results"). No such patterns existed when we tested these transcriptional regulators as hub genes. While *Mähler et al. (2017)* found an enrichment for transcription factors among hub genes in a single species network of *Populus tremula*,

our results in a multi-species network find the opposite pattern. Shifting the expression patterns of various genes and pathways can lead to phenotypic divergence among closely related species (*West-Eberhard, 1989*). This can be seen during later stages of flower development when distinct traits, such as red or purple flower color, can appear in different species due to shifting expression patterns within the anthocyanin biosynthetic pathway (*Smith & Rausher, 2011*). The connection between the apparent overabundance of transcriptional regulators in later stages of flower development and the rapid phenotypic divergence in floral form will need to be explored further.

Several notable peripheral nodes were identified for their potential involvement in important floral processes (Table S7). Three homologs were identified in the periphery of the Bud stage with potential involvement in the determination of zygmorphic (bilateral) floral symmetry: two homologs of divaricata (*DIV*; cluster13684_1rr.inclade1.ortho1.tre, cluster8585_1rr.inclade1.ortho1.tre; ME26 and ME0) and one homolog of dichotoma (*DICH*; cluster13245_1rr.inclade1.ortho1.tre; ME0). Bilateral symmetry has been proposed as a mechanism that facilitates pollen transfer by insects (*Galliot, Stuurman & Kuhlemeier, 2006*). Both *DIV* and *DICH* are TCP transcription factors that were first characterized in *Antirrhinum* (*Almeida, Rocheta & Galego, 1997*; *Luo et al., 1996*), and show localized expression to the ventral and dorsal portions of the petal, respectively. *Achimenes* flowers at the Bud stage are lacking or only just beginning to establish floral asymmetry and the relegation of both *DIV* and *DICH* to the network periphery may reflect this small role. The role of these genes has not been studied in the Gesneriaceae, but have been examined more recently in other non-model systems for their role in the evolution of pollination syndromes (*Zhang, Kramer & Davis, 2010*; *Preston, Martinez & Hileman, 2011*). Additionally, a homolog of the homeobox wuschel gene (*WUS*; cluster9273_1rr.inclade1.ortho1.tre; module ME0) was identified from the periphery of the D stage network. *WUS* was first characterized in *Arabidopsis* for its role in meristem cell maintenance (*Laux et al., 1996*) and its expression is localized only to the meristem (*Mayer et al., 1998*). Once the floral meristem has been initiated, *WUS* starts to be repressed by *AG* (hub gene, described above) (*Sun et al., 2009*; *Lohmann et al., 2001*). The relegation of *WUS* to the periphery in our networks at the D stage might then be expected given that it plays such a prominent role during early floral development and the vegetative to reproductive transition.

## Evolutionary rates are correlated to network connectivity

Increased protein evolutionary rates during rapid diversification has been suggested in other radiations of animals and plants (*Kapralov, Votintseva & Filatov, 2013*; *Brawand et al., 2014*; *Pease et al., 2016*). Our study is among the first to examine patterns of network evolution across multiple closely-related species of plants with multiple transitions in flower type and pollination syndrome. Hummingbirds are hypothesized to have had an important influence on diversification in Neotropical Gesneriaceae (*Roalson & Roberts, 2016*; *Serrano-Serrano et al., 2017b*), while the role of butterflies has not been explored. Modules correlated with hummingbird or butterfly pollination syndromes did not exhibit clear increases or decreases in evolutionary rates compared to other modules (Table S9). Detecting selection is subject to many factors, such as a gene's transcriptional abundance
and its importance within the protein interaction network (*Lemos et al., 2005*). For instance, we found that higher network connectivity was strongly associated with lower evolutionary rates across both networks in our study (Fig. 7; Table S9). This corroborates the recent results found in other eukaryotic systems (*Morandin et al., 2016*; *Masalia, Bewick & Burke, 2017*; *Josephs et al., 2017*; *Mähler et al., 2017*) while also showing the apparent strength of correlation between connectivity and evolutionary rate differs between studies and systems.

Network effects (such as the type of network analysis employed) and the expression levels of highly abundant genes are known to affect evolutionary rates, potentially confounding analysis of selection pressure (*Krylov et al., 2003*). We found that $d_N/d_S$ in early development may be dependent on both connectivity and expression levels, while this interactive effect was gone during later development (Table S9). As there are few highly connected genes (hubs, which are important determinants of the observed network structure), a random mutation would be less likely to affect such a gene. Changing the amino acid sequence of a hub gene could have multiple associated effects, some deleterious (*Hahn & Kern, 2005*; *Luisi et al., 2015*). In comparison, a random mutation would be more likely to affect a gene with lower connectivity, while also resulting in variants with fewer negative consequences. The hub genes have more direct interactions and are more likely to be essential and involved in more biological processes, placing them under greater constraint.

## Numerous network modules correlated to floral form

One primary use for gene co-expression network analyses is to identify modules of co-expressed genes whose overall expression (as measured by the eigengene) may correlate with different traits. These traits can be both qualitative or quantitative, such as social caste (*Morandin et al., 2016*), seed oil production (*Hu et al., 2016*), or anther development (*Hollender et al., 2014*). While we tested only qualitative traits here, future studies could also correlate eigengene expression to quantitative variables such as nectary size or sucrose concentration, since these appear to be another important component of pollination syndrome delimitation (*Perret et al., 2001*; *Katzer, Wessinger & Hileman, 2019*; *Vandelook et al., 2019*). Since our dataset was composed of samples from 12 species, we performed correlation analyses using a phylogenetic model to test whether module expression correlated to different flower colors, flower shapes, corolla spurs and pollination syndromes. Nearly half of our modules in both networks had evidence for an association with these floral traits (Figs. 5 and 6). There were no modules whose correlation to a pollination syndrome had increased eigengene expression in all species sharing that trait (Figs. 5 and 6). Instead many modules with correlation to a trait showed a phylogenetic pattern with a shared increased eigengene expression among the most closely related species. For example, module ME23 in the Bud stage network had a correlation to butterfly pollination and increased eigengene expression in *A. cettoana* and *A. longiflora*, both members of Clade 1 (Fig. 2). Our aim was to identify sets of co-expressed genes that would provide candidates for involvement in the development of these traits.

For instance, many of the candidate hubs we identified (see above) were found in modules that were correlated to one or multiple traits to which they might contribute, such as phantastica (*PHAN*; cluster11220_1rr.inclade1.ortho3.tre; module ME4, correlated to corolla spurs) and *DFR* (module ME48, correlated to hummingbird pollination and purple flowers) (Table S2) (*Waites et al., 1998*). Additionally, modules correlated to various floral traits were enriched for multiple genes that might be involved in the development of those traits (Table S8). Module ME53 was correlated to the corolla spurs found in *A. grandiflora* and *A. patens* in the D stage network and was enriched for many genes involved in cell division and cell growth (Fig. 6; Table S4). Other times, modules were correlated to a specific trait but the enriched GO terms did not provide a clear idea of how these co-expressed orthogroups may contribute (Tables S3 and S4). In our analyses, we considered relatively few traits and only focused on those that have traditionally been considered the most important for the differentiation of floral form in *Achimenes*, namely flower color, flower shape, and corolla spurs. Undoubtedly there are a myriad number of molecular and biochemical pathways that we have not considered here that underlie the development of these complex floral phenotypes. With the idea that networks represent complex systems of gene and protein interactions, it may be that these seemingly simple and generalized floral phenotypes are produced through much more complex means than many have previously considered.

## Strengths and limitations of network analyses in non-model organisms

Our analysis covers genes with orthologs found in most species, representing a set of evolutionarily conserved genes. While no reference genome currently exists for Neotropical Gesneriaceae, we constructed, annotated, and presented de novo transcriptomes for 10 *Achimenes* species as well as two outgroup species, *E. verticillata* and *G. cuneifolia*. Our approach enables functional genomic studies in this non-model system, but the potential for misassemblies may bias our downstream results (*Hornett & Wheat, 2012*). Using a gene tree-based approach to carefully identify orthogroups, we focused our attention on conserved gene regulatory machinery and not on taxonomically restricted genes. The latter may be involved in species-specific functions. Nevertheless, it is worth noting that many recent studies have indicated that transitions in floral form may be due to the shifting expression of genes found in conserved genetic pathways (*Wessinger & Rausher, 2015*). Taxonomically restricted genes are likely interacting with existing regulatory pathways, and the manner in which they integrate into the co-expression modules identified here will be a fascinating topic for future research. With extended sampling within *Achimenes* we could infer species-specific co-expression networks and perform more detailed cross-species comparisons. We expect that these taxonomically restricted genes may be poorly connected with the network, which may allow for increased diversification.

We characterized gene expression patterns by sequencing RNA across two timepoints during flower development. Previous gene expression experiments across three timepoints (early Bud, intermediate D, and late Pre-anthesis stages) in four *Achimenes* species suggested that the majority of differential gene expression occurred between the early Bud

stage and the intermediate D stage (*Roberts & Roalson, 2017*). We believe our approach allows us to capture dynamic transcriptional patterns during flower development for comparison across multiple species. We found that the co-expression networks for the Bud and D stages were largely preserved, despite there being a few stage specific modules (Table S5). However, this approach may greatly over-simplify the nature of transcriptional regulation in these species. Organ- and structure-specific expression studies may have greater power to detect floral form-specific differences, both in terms of the number of differentially expressed genes, and co-expression network structure (*Hollender et al., 2014*; *Suzuki et al., 2017*; *Shahan et al., 2018*). As a result, our data likely represents an underestimate of the true number of modules. Future studies could focus on the comparative analysis of expression patterns in specific organs, such as the petals, stigma, and stamens, and additional timepoints, which would provide greater functional insight into the evolution, development, and diversification of these organs.

Network analysis represents a more complex approach than differential expression analyses because it can capture system-level properties (*Langfelder & Horvath, 2008*). Most phenotypes involve interactions of proteins from diverse biochemical pathways. Although inferred modules do not necessarily correspond to entire biochemical pathways, or other components of cellular organization, the approach performs well in reconstructing the complexity of protein-protein interaction networks (*Allen et al., 2012*). In many systems this approach has succeeded in identifying candidate genes for various biological traits, for example floral organ development in strawberry (*Hollender et al., 2014*; *Shahan et al., 2018*) or carotenoid accumulation in carrots (*Iorizzo et al., 2016*). Our results demonstrate the difficulty in identifying gene clusters that underlie highly correlated floral traits, especially in non-model systems, as seen by the overlapping correlation of modules associated with flower shape and color (Figs. 5 and 6). It has been shown in many plant systems that floral traits tend to be selected together by pollinators (*Cuartas-Domínguez & Medel, 2010*; *Sletvold, Grindeland & Ågren, 2010*), making it challenging to separate correlated modules from causal modules. Co-expression networks are foundationally about correlations between genes (*Van Dam et al., 2018*), making them useful in identifying novel sets of genes that may be functionally relevant for trait of interest. Other approaches, such as quantitative trait locus (QTL) analyses may provide more promise in identifying distinct genes that are involved in the divergence of floral traits. QTL studies have already shown promise in identifying genetic loci that underlie pollination syndrome divergence in *Mimulus* (*Yuan et al., 2013*), *Penstemon* (*Wessinger, Hileman & Rausher, 2014*) and *Rhytidophyllum* (*Alexandre et al., 2015*).

We were able to identify numerous candidate orthogroups involved in the development of floral form in our non-model system. While no functional genomic data exists yet for *Achimenes*, our annotations are based on the manually curated SwissProt database. De novo assemblies provide important data for non-model organisms, but there can be biases introduced during assembly and annotation due to potentially missing genes (*Hornett & Wheat, 2012*). We attempted to limit this bias by focusing only on shared orthogroups found in the majority (at least six) of our sampled *Achimenes*. Applying this network-based approach to the diversification of flowers across multiple species in

*Achimenes*, we recovered distinct modules of co-expressed genes that may underlie a range of floral phenotypic traits tied to different pollination syndromes. Although our data come from whole flowers during development, and do not allow us to test specific hypotheses regarding whether particular genes have shifted expression location in the developing flower, they do suggest that there are conserved regulatory modules that may have been co-opted in convergent flower types multiple times. It will be interesting to apply network analyses to study floral diversification in additional lineages and examine how conserved or divergent the patterns are across angiosperms.

## CONCLUSIONS

Floral forms corresponding to bee, butterfly, and hummingbird pollination have evolved multiple times across the Neotropical plant genus *Achimenes*. Genome-wide gene expression estimates were taken from flowers in 12 species across two development stages in order to construct, analyze, and compare stage-specific gene co-expression networks. We hypothesized that numerous modules in each network would correlate to these pollination syndromes and that central genes in the network may be candidates for involvement in the development of important floral traits, such as flower color and shape. We found that nearly a third of modules were correlated to the pollination syndromes and many more were correlated to different flower colors and shapes. The outgroup species, *E. verticillata* (bee pollinated) and *G. cuneifolia* (hummingbird pollinated), displayed correlation patterns similar to the ingroup species that shared floral traits, suggesting that some co-expression patterns might be shared across evolutionary distances. Several of the hub genes in the networks were homologs of the anthocyanin and carotenoid biosynthetic pathways, important for the production of floral pigments that attract different pollinators. A negative relationship between network connectivity and $d_N/d_S$ corroborates the findings in model systems that more centrally located nodes (likely hubs) are under increased evolutionary constraint. We found that the less connected genes (peripheral) are under more relaxed constraints and contained numerous transcriptional regulators. Our results demonstrate the utility of applying co-expression network analyses in non-model plant lineages to begin identifying important modules and pathways that will be useful starting points for additional analyses into the evolution and development of floral form.

## ACKNOWLEDGEMENTS

We thank Joanna Kelley, Amit Dhingra and Andrew McCubbin for helpful comments that improved this work. Sequencing was performed at the Genomics Core Lab at Washington State University, Spokane, and the Genomics Sequencing and Analysis Facility at the University of Texas, Austin.

### Funding

This work was supported by a National Science Foundation Doctoral Dissertation Improvement Grant (DEB 1601003), the Elvin McDonald Research Endowment Fund

from The Gesneriad Society, and the Global Plant Sciences Initiative Fellowship from Washington State University. The funders had no role in study design, data collection and analysis, decision to publish, or preparation of the manuscript.

### Grant Disclosures
The following grant information was disclosed by the authors:
National Science Foundation Doctoral Dissertation Improvement: DEB 1601003.
Gesneriad Society.
Washington State University.

### Competing Interests
The authors declare that they have no competing interests.

### Author Contributions
- Wade R. Roberts conceived and designed the experiments, performed the experiments, analyzed the data, prepared figures and/or tables, authored or reviewed drafts of the paper, and approved the final draft.
- Eric H. Roalson conceived and designed the experiments, analyzed the data, authored or reviewed drafts of the paper, and approved the final draft.

### Data Availability
Raw sequencing data is available at NCBI SRA, BioProject numbers: PRJNA435759 and PRJNA340450.

The data and code are available at: Wade R. Roberts & Eric H. Roalson (2020). Co-expression clustering across flower development identifies modules for diverse floral forms in Achimenes (Gesneriaceae) (Data set). Zenodo. DOI 10.5281/zenodo.3517231.

### Supplemental Information
Supplemental information for this article can be found online at http://dx.doi.org/10.7717/peerj.8778#supplemental-information.

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
