# Peer review of "Co-expression clustering across flower development identifies modules for diverse floral forms in Achimenes (Gesneriaceae)"

_PeerJ, doi:10.7717/peerj.8778_

## Round 0.1 · original submission · Major Revisions

Many issues regarding methods, clarity of Introduction and the way the results are presented were raised by the three reviewers and they should be considered carefully for the next round of reviews. You need to explain how you dealt with every comment clearly. Among the crucial arguments are to introduce information on the pollination biology of the studied species, the lack of specific developmental landmarks for the floral stages selected, the availability of raw data and clarification in Discussion on the delimitation of co-expression networks. In addition one of the reviewers suggested the paper should reach a professional English.

Reviewer 1 ·

Basic reporting

Language comments
The manuscript is written in ambiguous landuage in some part, eg. P46-47, P51, P58-59 ……and so on. Writing skills must be improved before acceptance.
Introduction
1. The introduction lack the literature about the specific pollinators of 10 species.
2. The correct citation format should be (Pickrell et al. 2010; Raherison et al. 2015) in P59, so as the follows.

Experimental design

1. The flower forms of 10 species should be described more clearly, such as the colors and types of each species.
2. The bud stage and D stage should be describe by the paraffin sections to clarify the difference of them. And you can also supply the pictures under the dissecting microscope.

Validity of the findings

1. P377-378 More figures are needed to prove it or summarize it.
2. P384-385 Please provide evidence to support this sentence.
3. P389-390 It is inconsistent with the figures, please check it. \
4. There are only few hub genes are mentioned in the results and more hub genes relevant the flower development should be discussed. And the expression of hub and candidate genes should be vertificated by the qRT-PCR.

Additional comments

I am glad to have the opportunity to evaluate your article titled “Co-expression clustering across flower development identifies modules and candidate genes for diverse floral forms in Achimenes (Gesneriaceae)”. In general, it is a reasonable article as you built a transcriptome database using 10 species of Achemenes. And Co-expression clustering and relationship between network connectivity and evolutionary rates (dN/dS) were done to identify modules and candidate genes for diverse floral forms in Achemenes. Nevertheless, there are a few issues in the MS need to be clarified.

Reviewer 2 ·

Basic reporting

English is of sufficient quality, references are updated and data is shared in public repositories.

Experimental design

No comment

Validity of the findings

The validity of the findings depends on a better description of the developmental stages sampled. The section "Hub nodes contain important candidates for floral form" requires a more comprehensive discussion as in its current form only an ad hoc example is used to assess their contribution to floral form but extensive expression and functional literature is available for each of the genes reported.

Additional comments

PeerJ
Co-expression clustering across flower development identifies modules and candidate genes for diverse floral forms in Achimenes (Gesneriaceae)

The authors generate floral transcriptomes from 10 species of Achimenes in two floral developmental stages. Their goal is to identify the genes, gene clusters and possibly gene networks underlying floral shape and other features responsible for shifts in pollination syndromes. The strengths of the manuscript lay on the ambitious computational approach undertaken and the generation of a large dataset for comparative transcriptomics in this non-model taxon. The weaknesses are the lack of specific developmental landmarks for the floral stages selected, and the lack of candidate genes at the end of the analyses that are never presented but that are suggested from the title. In addition, my recommendation is to rescue the biological significance of the results. Currently much of the descriptions are concentrated in module description (i.e. (‘brown4’, ‘coral2’, ‘floralwhite’, ‘lavenderblush3’, ‘lightcyan1’, ‘lightsteelblue1’, ‘maroon’, ‘mediumpurple3’) and the real floral features are harder to pick up from these descriptions. Can clusters at least be numbered instead of described by colors?.

Introduction

Line 54. “flowers may be under constant and strong selective pressure”. Strong selective pressure might be associated with the whole plant, reflecting adaptation to abiotic or biotic selective pressures. Color variation is also related to drought stress, soil changes, herbivory, and other factors that vary spatially.

Line 61. “The transcriptome is highly dynamic”. Which transcriptome?. I think the sentence can be removed as it corresponds to a general statement on transcriptomes.

Line 72. “It is now recognized that most genes act as members of biological pathways or of co-regulated modules”. There are other recent studies that could be cited (2014-2018), with emphasis in plants co-regulated modules, that explained even more complexes interaction than just co-expressed pairs of genes as suggested in Stuart et al. 2003. For example, Ma et al., 2018. Co-expression gene network analysis and functional module identification in Bamboo growth and development.

Line 79. Syndromes are in parenthesis and the sentence really refers to pollinators. Please revise.

Line 80. Figure 1. The authors included an informative and clear figure showing the phylogenetic relationships in Achimenes. However, despite the fact that the figure title indicates “floral form” such trait is not easy to identify from the pictures of A. longiflora, A. cettoana, A. grandiflora, etc. New pictures perhaps similar to Figure 1B of Roberts and Roalson, 2017 may be necessary.

Line 83. “Repeated origins and transitions between these pollination syndromes have been hypothesized in Achimenes”. Please include a citation.

Line 100. “to identify candidate genes and pathways that may be associated with the repeated evolution of floral forms underlying pollinator specificity”. Only form was significant to the pollinator specificity? What happens with the color? Any relevant or significative finding linked to this trait?


Materials and Methods

Line 114. “an intermediate stage before anthesis”. All stages need to be described in terms of developmental timing and landmarks. Size does not suffice to make stages comparable. This is one of the weakest points of the manuscript as later on the hubs go on to identify floral traits (for instance spurs) associated modules and it is unclear if they are already beginning to form in the sampled points.

Line 145. “Grabherr et al. 2011; Haas et al. 2013”

Line 178. This information: “Rows corresponded to orthogroups and columns corresponded to individual samples” is associated with a Table or Suppl. Material?

Line 193. What was the threshold used to determine the edges of the corresponding TOM similarity results?


Results

Line 274. Due to the information included in this section: “Our focus in our comparative study was on the presence of conserved orthologs, and not on the presence of taxonomically restricted genes, which are not likely preserved across species. We subsequently identified 83595 orthologous clusters (orthogroups), each containing at least 2 species”. I suggest changing the title “Transcriptome assembly” to “Transcriptome assembly and orthogroups identification” or move this part to the beginning of the next section “Constructing the co-expression network for 12 species”.

Line 281. “De novo assembly enables functional genomics studies in non-model systems without a reference genome, but has the potential for misassemblies (Hornett and Wheat 2012) and subsequent downstream biases”. This paragraph seems to be part of the discussion and it suggests the approach is not free from bias which is somehow negative.

Line 282. Although, the misassembly is a possible consequence of the De novo assembly process; Hornett and Wheat (2012), also suggest a level of bias in the genes that can be annotated in the resulting transcriptome, were genes might be potentially missing during the analyses.

Line 309. It might be interesting if the genes inside these non-preserved modules could be assigned to other functional categories, such as transcription factors or constitutive genes.

Line 316. Figure 2: I suggest split this figure in two different figures, one for the current A and B (co-expression network analyses) and put the correspondence of Bud and D stage modules (C) in a different figure. As it is the shared modules between the Bud and D stage, and their specific number of genes are hard to read.

Line 346. In “Modules associated with floral traits” it is stated that “WGCNA does not use trait information when inferring modules, so modules that correlated strongly with any floral trait were inferred without a priori knowledge”. However, it is interesting only few species are correlated to each module for a specific trait. Please explain the phylogenetic mixed model, and how was that used to avoid or control the phylogenetic bias and their possible influence in the association that was done with the floral traits.

Line 352. Presence/absence of spurs is unclear in Fig 2. Please add the information missing.

Line 369. Please include information about specific functional categories, regarding the genes in each module associated with the pollination syndromes, flower color, etc.


Discussion


Line 442. In the eigengene expression results: modules without significant association (> 1.0 or < -1.0), for example, “bisque4”, “honeydew1”, and “ivory”, in Figures 3 and 4, might suggest these are housekeeping genes?

Also, why are these rows lacking colors for “strongly positive”, “weakly positive”, “strongly negative” or “weakly negative”, when same rows show species with significant values of eigengene expression (>1.0 or <-1.0), in the two right columns in Figures 3 and 4?

Line 477. This is important and might be interesting to test this approach of co-expression network analyses, using quantitative variables in the “eigengene” correlation. For example: the presence of nectar (nectar volume, sucrose concentration, etc). [See: Vandelook et al., 2019. Nectar traits differ between pollination syndromes in Balsaminaceae. Annals of Botany, Vol 124, Issue 2, 269-279.]

Line 493. The section "Hub nodes contain important candidates for floral form" requires a more comprehensive discussion as in its current form only an ad hoc example is used to assess their contribution to floral form but extensive expression and functional literature is available for each of the genes reported. At this point this section requires a more rigorous literature revision.

Line 496. Genes identified (AG, TCP4, GLO, etc.) are only discussed selectively from some species, but the choice for discussing in such particular taxa is unclear. Sometimes the reference is Arabidopsis, sometimes Aquilegia, sometimes Antirrhinum. Authors need to explain why specific roles are highlighted and not others.

Line 567. There is no year of publication for Hahn and Kern (2004) in the references.

Line 583. Number of modules or the percentage of modules with correlation to any specific trait (shape, color, etc.) at the Bud and D stages is not shown. Can authors report the number of sets of co-expressed genes consistent between the predicted trait and the actual trait of the flowers in each species or species clade?.

Line 585. The module “darkturquoise” in the Bud and D stages, shows a correlation with the expected pollination syndrome (butterfly) and color (purple). However, the eigengene expression is strongly positive in both traits for A. grandiflora (AG) and A. patens (AP) at the D stage, and A. acettoana (AC) and A. longiflora (AL) at the Bud stage. Is there any possible explanation for the differences regarding the evaluated developmental stage?

Line 594. The module “palevioletred3” was correlated to corolla spurs in the D stage network in the species, A. grandiflora (AG), A. patens (AP), and E. verticillate (EV). However it is not clear if spurs have been identified in these species and when does it develop in these species?

Line 647. Please include Suzuki et al. 2018 in references. In references was only included Suzuki et al. 2017 (cited in line 635).

Line 661. Please include a little discussion or conclusion about the results obtained for the two outgroup species that were included in this study (E. verticillata and G. cuneifolia).

Line 665. The sentence “It will be interesting to apply network analyses to study floral diversification in additional lineages and examine how conserved or divergent the patterns are across angiosperms” is an over statement and this will likely change group by group. Please remove and replace with a conclusion more specific to Gesneriaceae flower evolution.

·

Basic reporting

Everything is fine except for the availability of the raw and processed data that is not available at this point (see general comments below).

Experimental design

Fine.

Validity of the findings

All underlying data not provided yet. The rest is fine.

Additional comments

This manuscript builds co-expression gene networks from RNAseq data across 10 species of Achimenes (Gesneriaceae) plus two outgroups with contrasting pollination strategies (bees, hummingbirds, and butterflies) to investigate the genetic architecture of flower differentiation. The study seems novel in that I have not seen such an approach applied to investigate the genetic of flower shape across species. Even if few genes are discussed specifically—the attention is given more to the modules—the results are interesting and relevant. The link between the ratio of synonymous to nonsynonymous nucleotide substitutions and the position of genes in the networks is also very interesting to my opinion and tends to validate the whole approach. Overall, the results are certainly of great interest to all scientists that are interested in the genetic determination of flower shape and color and its impact on floral diversity.

Although I don’t have much personal experience with co-expression modules, all analyses conducted in the manuscript seemed very robust and performed adequately (but see comments below). I also found the discussion pertinent and well written.

Most of my comments are rather minor, but I have one that could be slightly more important. It has to do with the delimitation of the co-expression networks as well as their interpretation and discussion. I was surprised to see that several modules that should be independent by definition tend to correlate with the same floral traits. Why is this so? Is it because the threshold of some analyses was too low (either for the module construction or for the correlation analyses)? For instance, how were the settings for the construction of the modules determined? I would be curious to know what happens if you lower the threshold for including genes within a module.

One possibility to explain these results would be that most traits are highly correlated and that it is difficult (maybe impossible) to distinguish the correlations between modules and traits that are due to causation from those that are correlational. Indeed, the fact that numerous traits tend to be selected together by pollinators and this in a convergent manner in different evolutionary lineages could make it impossible to make a causal link between expression modules and the traits they affect. A discussion of these issues seems important in the section on the limitation of co-expression networks in the discussion. This is perhaps where other approaches like QTL analysis becomes very handy to distinguish the effect of genes on specific floral traits. Most QTL studies to data have shown that distinct genes are involved in color, floral shape, nectar, etc. even if they show a strong evolutionary correlation because they are all selected in a convergent manner by pollinators. It seems highly likely that comparative approaches between species will often fail to precisely define the genetic architecture of the traits selected concomitantly and rapidly along specific branches of the trees. This doesn't minimize the conclusions of this study, but this limit should be acknowledged.

Also, I noticed that the authors have decided to keep only the genes that had a positive correlation in expression in the modules (type="signed"). I think this should be justified. Actually, it might explain why different modules are correlated with the same traits (at least partially). Again, I am not an expert in such analyses, but it would have seemed logical to me to keep in a module all the genes that are strongly correlated, either positively or negatively.

Minor comments:

1. Why keeping orthogroups that are common to the bud and D stages? It seems likely that the genes expressed at these two stages are quite different. Since modules are built separately for the two developmental stages, why not keep distinct sets of genes for each of them? Is it to facilitate comparison of modules between developmental stages ?
2. Line 454: be more precise than “relatively few genes”. This can have different meanings for different persons. Also, although moderate numbers of genes are found to be different involve in floral changes between species (QTL studies), many more genes are involved in flower development in each species.
3. The first paragraph of the intro lacks a main trend. Presently looks like different sentences stitched together.
4. It would be important that all the data was made available following the manuscript acceptance, including genes info, GO terms, modules to which they belong, expression data, phylogenies, tables for analyses, etc. Presently, none of these are available and nothing is said in the manuscript of where they will be archived.

---

## Round 0.2 · accepted · Accept

I appreciate your effort for carefully taking into consideration suggestions, comments and issues raised by the three reviewers. Thank you for your offer of uploading raw data and code into a permanent repository.